# Spatially Enriched Paralog Rearrangements Argue Functionally Diverse Ribosomes Arise during Cold Acclimation in Arabidopsis

**DOI:** 10.3390/ijms22116160

**Published:** 2021-06-07

**Authors:** Federico Martinez-Seidel, Olga Beine-Golovchuk, Yin-Chen Hsieh, Kheloud El Eshraky, Michal Gorka, Bo-Eng Cheong, Erika V. Jimenez-Posada, Dirk Walther, Aleksandra Skirycz, Ute Roessner, Joachim Kopka, Alexandre Augusto Pereira Firmino

**Affiliations:** 1Willmitzer Department, Max-Planck-Institute of Molecular Plant Physiology, 14476 Potsdam-Golm, Germany; olga.beine@bzh.uni-heidelberg.de (O.B.-G.); hsieh.y.chen@uit.no (Y.-C.H.); eshraky@mpimp-golm.mpg.de (K.E.E.); michal.gorka@celonpharma.com (M.G.); becheong@ums.edu.my (B.-E.C.); walther@mpimp-golm.mpg.de (D.W.); skirycz@mpimp-golm.mpg.de (A.S.); kopka@mpimp-golm.mpg.de (J.K.); firmino@mpimp-golm.mpg.de (A.A.P.F.); 2School of BioSciences, University of Melbourne, Parkville, VIC 3010, Australia; u.roessner@unimelb.edu.au; 3Heidelberg University, Biochemie-Zentrum, Nuclear Pore Complex and Ribosome Assembly, 69120 Heidelberg, Germany; 4Institute for Arctic and Marine Biology, UiT Arctic University of Norway, 9037 Tromsø, Norway; 5Biotechnology Research Institute, Universiti Malaysia Sabah, Jalan UMS, 88400 Kota Kinabalu, Malaysia; 6Grupo de Biotecnología-Productos Naturales, Universidad Tecnológica de Pereira, Pereira 660003, Colombia; evjimenez@utp.edu.co; 7Emerging Infectious Diseases and Tropical Medicine Research Group—Sci-Help, Pereira 660009, Colombia

**Keywords:** functional heterogeneity, paralog subfunctionalization, remodeling, ribosomal code, ribosome-associated proteins, ribosome biogenesis, stress-specialized ribosomes, substoichiometry

## Abstract

Ribosome biogenesis is essential for plants to successfully acclimate to low temperature. Without dedicated steps supervising the 60S large subunits (LSUs) maturation in the cytosol, e.g., Rei-like (REIL) factors, plants fail to accumulate dry weight and fail to grow at suboptimal low temperatures. Around REIL, the final 60S cytosolic maturation steps include proofreading and assembly of functional ribosomal centers such as the polypeptide exit tunnel and the P-Stalk, respectively. In consequence, these ribosomal substructures and their assembly, especially during low temperatures, might be changed and provoke the need for dedicated quality controls. To test this, we blocked ribosome maturation during cold acclimation using two independent *reil* double mutant genotypes and tested changes in their ribosomal proteomes. Additionally, we normalized our mutant datasets using as a blank the cold responsiveness of a wild-type Arabidopsis genotype. This allowed us to neglect any *reil*-specific effects that may happen due to the presence or absence of the factor during LSU cytosolic maturation, thus allowing us to test for cold-induced changes that happen in the early nucleolar biogenesis. As a result, we report that cold acclimation triggers a reprogramming in the structural ribosomal proteome. The reprogramming alters the abundance of specific RP families and/or paralogs in non-translational LSU and translational polysome fractions, a phenomenon known as substoichiometry. Next, we tested whether the cold-substoichiometry was spatially confined to specific regions of the complex. In terms of RP proteoforms, we report that remodeling of ribosomes after a cold stimulus is significantly constrained to the polypeptide exit tunnel (PET), i.e., REIL factor binding and functional site. In terms of RP transcripts, cold acclimation induces changes in RP families or paralogs that are significantly constrained to the P-Stalk and the ribosomal head. The three modulated substructures represent possible targets of mechanisms that may constrain translation by controlled ribosome heterogeneity. We propose that non-random ribosome heterogeneity controlled by specialized biogenesis mechanisms may contribute to a preferential or ultimately even rigorous selection of transcripts needed for rapid proteome shifts and successful acclimation.

## 1. Introduction

Due to the sequential nature of ribosome biogenesis, groups of ribosome-associated proteins (RAPs), including structural ribosomal proteins (RPs), are transiently or permanently mounted into the pre-ribosomal complexes [1,2,3]. Transiently binding RAPs include ribosome biogenesis factors (RBFs), which monitor and assist assembly, processing, and maturation of ribosomes. Permanently binding proteins include RPs, which are part of mature translationally competent ribosomes. RPs are encoded by several paralogs in plants [4], thus compared to other eukaryotes, the ribosomal proteome of plants is highly diverse [5]. The high number of RP paralogs creates a combinatorial explosion of theoretically possible heterogeneous ribosomes. What may be seen as random and likely redundant paralog heterogeneity might have been directed towards functional divergence in the course of plant evolution, by paralog sub- and/or neofunctionalization [6,7]. Differential use of RPs or paralogs to build specialized ribosomes requires the complexes to be newly synthesized or remodeled after an environmental stimulus. *De novo* synthesized ribosomes with altered paralog compositions first appear in non-translational complexes, i.e., free 40S (SSU) and 60S (LSU) subunit fractions, before these are assembled into translating monosomes and polysomes. Subunits are assembled during the early nucleolar biogenesis steps where different factors catalyze defined steps in a highly ordered assembly line. Groups of RPs are mounted in concert following systematic directives. Thus, stress cues that compromise specific biogenesis steps may trigger regional rearrangements in spatially adjacent RP groups. Additionally, even though the assembly of RPs or individual paralogs into ribosome complexes may depend purely on the relative abundance of available, not yet assembled proteins at the assembly sites, structural constraints may exist that prefer concerted mounting of combinations of RPs or individual paralogs that are structurally or functionally adjusted to each other. These two possibilities imply that upon external stimuli groups of RPs may be jointly modulated and constitute an altered spatial region of the ribosome with a defined function. Thus, stress-remodeled ribosomal fractions are likely to be substoichiometric [8] compared to canonical RP compositions. A substoichiometric stress-specific ribosome population may carry structural features that influence the translational status of transcripts to achieve selective translation and rapid proteome shifts, a concept termed the ribosomal code [9,10,11].

Already in simpler eukaryotic ribosomal proteomes, as compared to that of plants, selective translation can be achieved by differential use of RPs or RP paralogs [12,13]. In fact, however, eukaryotes select transcripts for translation through several distinct mechanisms [14,15]. These mechanisms usually rely on either transcript features, altered ribosome structures, and/or RAPs. In plants, for example, there are global translational responses to abiotic and biotic stresses that imply differential association of transcripts with ribosomes [16,17,18]. A prominent example is heat stress, which causes extensive repression of mRNA translation in both *Oryza sativa* and *Arabidopsis thaliana* [19,20]. Besides global regulatory mechanisms, there are targeted translation constraints triggered by exogenous stimuli. A first class of targeted mechanisms relies on transcript features. For instance, plants undergoing oxygen deprivation favor translation of mRNAs that have low GC content in their 5′ UTR [21,22,23]. Conversely, transcripts with GC rich 5′ UTRs are translationally repressed during dark and hypoxia conditions [24]. Transcripts with a cis-element, TAGGGTTT, in their 5′ UTR feature longer half-lives and shorter cDNAs, which correlate with increased translation rates in Arabidopsis [25]. A second class of targeted mechanism relies on altered ribosome structures to impose translational controls. For example, altered RP compositions endow ribosomes with selectivity towards sub-pools of mRNAs in mammalian cells [12]. The selectivity is partly achieved by recognition of transcript IRES- elements (i.e., internal ribosome entry sites) that influence translation initiation in a cap-independent manner. In *Caenorhabditis elegans*, methylation of an 18S rRNA adenosine enhances selective binding and translation of mRNA subsets [26]. In humans, poxviruses are able to remodel the host ribosome by phosphorylation of a plant-like charged RACK1 loop, mimicking a plant-like state that favors translation of viral RNA [27]. Finally, a third class of targeted mechanisms actually relies on transient ribosome-associated factors to achieve selective translation. For instance, the yeast homolog of RACK1, ASC1, promotes efficient translation of short mRNAs [28]. mRNAs that are translationally favored by RACK1 preferentially associate with the initiation complex composed of eIF4E, eIF4G, and Pab1 [28]. Similarly, in plants, the phosphorylation event of eukaryotic initiation factor 2α by GCN2 mediates cadmium stress and confers selection capabilities for translated transcripts [29]. Non-canonical RAPs such as kinases can even connect translation to diverse cellular processes [30]. RAP-mediated selective translation generally involves the modification of RPs, which, when linked to specialized TIFs [31], serves the purpose of shaping the translated proteome. Evidently, heterogeneity of the ribosome structure alone does not suffice to enable a ribosomal code. Rather, interactions between transcript recruiting mechanisms, specialized ribosomal populations, and associated ribosomal factors must be considered.

The struggle of plants to cope with cold acclimation is especially critical due to their sessile habit and has emerged as an exemplary case where many of the mentioned translational mechanisms appear to act independently or synergistically to achieve successful acclimation [32,33]. At the onset of cold acclimation, plants halt growth for a quiescent period of ~7 days, during which a global reprogramming of the transcriptome occurs [34]. The reprogramming is concomitant to transcript splicing [35], ribosome and spliceosome component accumulation, and proteome shifts [36]. At the transcriptome level, genes that have been termed as “early” and “late cold responsive” peak in altered expression levels at 2 and 12 h after the initial stimulus [37]. These responsive categories include protein coding genes from the cytosolic ribosomal proteome, which appear to retain their altered expression patterns beyond the initial 24 h of acclimation [34,38]. In our current study, we used the same cold stress treatments previously characterized to trigger a classical plant cold response [34,39] in order to delve into the more specific aspect of translational reprogramming. Consequently, we report that structural RP coding genes peak in altered expression at the end of the quiescent period (~7 d), whereas biogenesis and other ribosome-associated factors peak in altered expression patterns within the first day of acclimation. Thus, triggering an early ribosome biogenesis cold response may be extremely important and induce a later transcriptional response in RPs. In agreement with the indispensability of cold-induced ribosome assembly, RBFs such as REIL proteins are essential to activate biogenesis only during cold in both yeast and in Arabidopsis [34,38,39,40,41,42,43]. These temperature-specialized proteins mediate the final LSU subunit maturation events in the cytoplasm, which happen concomitantly with the polypeptide exit tunnel (PET) quality control and the P-Stalk assembly [44,45]. PET assembly occurs much earlier in the nucleolus when pre-ribosomal particles are slowly processed and crafted [46]. Thus, disrupting PET assembly in the early biogenesis steps might cause a debilitated tunnel structure [47] and cause the need for a dedicated proof reading step catalyzed by REIL later on in the cytoplasm. After testing this example, we propose that plant ribosome heterogeneity is non-random and is likely controlled by specialized mechanisms during biogenesis that preferentially assemble groups of RPs or RP paralogs in a concerted process. This creates structurally diversified ribosome subpopulations. Specialized functions of these ribosome subpopulations may be executed and assisted by respective specialized RAPs that target all ribosome functions from translation initiation and elongation to termination.

In our current study, we investigate whether suboptimal low temperatures induce protein-heterogeneity in ribosomal complexes, and test whether the induced changes are confined to specific regions of the ribosome structure. To do that, we conjugated three independent genotypes to produce a proteomic dataset that statistically excluded any induced changes that happen to LSU ribosomal particles after the cytosolic maturation steps. We achieved that by using two independent double knock out mutants of the REIL alleles in conjunction with the Arabidopsis wild type Col-0. Then in order to identify cold- specific effects on the ribosomal proteome, we analyzed altered transcript levels and RP abundances after sucrose density sedimentation. Our analyses center on free LSU fractions since REIL genotypes allowed us to block their maturation during cold. We then explored co-localization of temperature-responsive RPs and paralogs within the ribosome body and found changed ribosomal regions of spatially related RPs. These RPs constitute significantly modulated regions whose entities feature transcripts or proteoforms that are differentially accumulated during cold. We discuss potential functions of the altered ribosome substructures at the onset of cold acclimation taking into account concomitant cold changes. In summary, our work provides evidence of stress-inducible and spatially constrained heterogeneity arising in plant ribosomes that is likely non-random and therefore functional.

## 2. Results

Plant roots contain three types of ribosomes: mitochondrial, plastid, and cytoplasmic. Chloroplasts are largely absent from roots, but non-green plastids are present. These plastids are required for root function [48] but generally do not perform photosynthesis and are present in lower amounts than leaf chloroplasts. Thus, ribosome preparations of root tissue allowed us to obtain larger relative amounts of cytosolic ribosomes than preparations from similar amounts of shoots. Hydroponic cultures, as those described in Firmino et al., 2020 [49] and Erban et al., 2020 [50], allowed rapid harvesting of root tissue with minimal perturbations. Hydroponic glass pots for Arabidopsis are small, making them optimal for highly replicated use within growth chambers set to varied environmental conditions, such as different temperature regimes.

Conventional plant cold stress experiments are performed at 4 °C, e.g., Ashraf and Rahman, 2019 [51]. The first and direct target of cold stress is the shoot, with root cold stress being more complex in terms of variables. The soil buffers temperature changes and causes a top-to-deeper soil temperature gradient, which typically increases with soil depth [52]. Hence, the soil delays corresponding temperature changes of the root system. By choosing a hydroponic growth system, we minimized temperature gradients and the delay of temperature changes between root and shoots. Thus, the typical 4 °C cold stress, which would affect shoots and then be buffered to the roots by soil layers, was translated into a 10 °C stimulus applied directly to the roots. Consequently, we ratified 10 °C as a legitimate meteorological condition within the temperature ranges that Arabidopsis roots face during natural autumnal or spring low temperature acclimation.

### 2.1. Early Temperature Acclimation Effects on Plant Growth

Physiological effects of plant cold acclimation include slower growth rates and increased water contents [53,54]. We estimated that during the first seven days of cold acclimation following a shift in the middle of the light cycle from 20 °C to 10 °C, *Arabidopsis thaliana* at vegetative stage 1.10 [55] halts growth and development in terms of leaf number, rosette area [34], and, according to our current study, dry weight accumulation (Figure 1). Plants reared and maintained at 20 °C confirm that without a temperature shift, plants significantly accumulate dry weight (Figure 1). Growth arrest after a temperature shift is not a specific cold response. Plants shifted to 30 °C halted dry weight accumulation as well (Figure 1). Plants exposed to 10 °C or 30 °C for longer periods than 7 days typically resume growth and develop inflorescences, which is why these ranges are regarded as suboptimal temperatures for plants. These observations imply that halted or reduced dry weight accumulation is a shared early and transitory response of Arabidopsis to suboptimal temperature shifts.

Plants have a tissue- and developmental stage-specific expression of RPs and RP paralogs [56,57,58]. Similarly, there may not be an invariant standard ribosome population across all environmental conditions. We argue that ribosomal-related plant physiology during temperature acclimation periods of induced growth arrest (Figure 1) is not dormant [36]. Instead, Arabidopsis could accumulate, for instance, specialized ribosomes in order to cope with prolonged temperature changes. In this study, we focused our transcriptomic and shotgun proteomic analyses on the rewiring of the translational apparatus and attempt to reveal RP or RP paralog changes triggered by temperature acclimation during early biogenesis. Our analyses stay within the vegetative growth phase and precede the *de novo* synthesis of ribosome piles that is inevitably associated with resumed growth of new tissue at suboptimal temperature. Instead, we analyze bulk root tissue that pre-formed at optimized temperature and subsequently acclimated to reduced temperature.

### 2.2. Cytosolic Ribosomal Transcriptome Reprogramming

We tested changes of transcript abundances in whole root systems immediately before the 10 °C cold shift and at 1 day or 7 days into cold acclimation (Appendix A). We initially included probe hybridization data of all potential RPs and RAPs that were reported earlier (Appendix A of Beine-Golovchuk and co-authors, 2018). Of these probes, 329 and 82 indicated changed expression of RAPs only after 1 day and 7 days of cold, respectively, at a false discovery rate (FDR) <0.05. On the other hand, 153 differentially hybridizing probes were shared among the time points (Appendix A). General expression trends over time were evident among the differentially hybridizing RP and RAP probes. Four clusters emerged that indicated a major cold-induced reprogramming of the ribosomal and associated translation-related transcriptome (Appendix A).

Subsequently, we focused our analyses on the cytosolic RP genes in an attempt to reveal cold-induced reprogramming of the cytosolic ribosomal proteome (RP) transcriptome (based on Appendix A from Martinez-Seidel et al., 2020). A considerable part of the 242 structural cytosolic RP genes that were represented by partially redundant 310 gene probes changed expression upon cold shift and contributed to the overall transcriptional reprogramming of translation-related genes. Of the RP probes indicated, 35 (11.3%) and 75 (24.2%) indicated changed gene expression after 1 day and 7 days of cold, respectively, at *p* < 0.05. A total of 15 (4.8%) differentially hybridizing probes were shared among the time points (Appendix A). Four expression trends of cytosolic RP genes were apparent among the significantly cold responsive transcripts of both the 40S SSU and the 60S LSU (Figure 2A,B and Appendix A). Additionally, a fifth group of non-cold-responsive probes indicated that 66 of 100 SSU transcripts and 86 of 142 of LSU transcripts remained unchanged (Appendix A). Among the four groups of responsive genes (Appendix A), part of the transcripts increased early after cold shift and continued to increase (Cluster 4). A second group decreased inversely (Cluster 3). A third cluster of RP transcripts increased late at 7 days after cold shift (Cluster 2). Finally, a fourth cluster of RP transcripts increased transiently at 1 day after cold shift and returned to initial expression levels (Cluster 1). A 40S group of responsive genes showed an opposite behavior, decreasing transiently at 1 day after cold shift and increasing again after 7 days. In summary, the transcriptional response of RAPs peaked at 1 day after cold shift (Appendix A), while only less than 1/3 of the responsive RPs belonged to this second group of early cold responsive genes [37]. The majority of structural RP transcripts are responsive at 7 days of cold acclimation (Appendix A).

Subdividing our expression data by RP family indicated that some paralogs within single gene families exhibited similar responses whereas other paralogs responded differentially, e.g., by early maintained increases and inverse early maintained decreases (Figure 2C). To simplify subsequent analyses, we subdivided the transcript data according to time after cold shift. Considering the two time points separately, we encountered three types of differential expression patterns among paralogs of the cytosolic ribosome gene families and mapped these trends onto a 3D rendering of the 80S wheat translating monosome [59] (Figure 3). The differential expression patterns were: (1) only significant increase of one or multiple paralogs within one family (Figure 2D top and Figure 3, purple color-code). (2) Only significant decrease of one or multiple paralogs of a gene family (Figure 2D bottom and Figure 3, yellow color-code), and (3) at least two significantly inverse temperature regulated paralogs within a RP family (Figure 2D and Figure 3—black color code). Cases where only specific paralogs of a cytosolic RP family have a differential expression, but other members of the same family do not share or even express the inverse of this response, may be interpreted as an indication of a paralog exchange during temperature acclimation.

Inverse transcriptional responses of RP paralogs are more abundant among the class of late responding transcripts (Figure 3—black color-code) at day 7. Upon visual inspection, we found that inversely changed paralogs appear spatially localized and are apparently not randomly scattered throughout the ribosome. Inversely temperature regulated paralog transcripts within an RP family may be seen as a strong indication of subfunctionalized paralogs, which are exchanged in ribosome pools according to changes of ambient temperature. A first example from our data set is the P-stalk that experiences an initial paralog-specific increase of gene expression at day 1, followed by an inverse regulatory pattern at day 7 after cold shift. A similar observation is apparent among protein-coding genes that constitute or are in proximity to the polypeptide exit tunnel (PET). Such spatial constraints may further indicate functional relevance of temperature-induced paralog exchanges in ribosome populations. Only a subset of RP transcripts were differentially regulated during cold. The identity of those transcripts as well as the outcome of a statistical test tailored to evaluate their spatial relationship in the ribosome structure are found in the results Section 2.6, “Spatially Constrained Cold Triggered Ribosome Heterogeneity”.

### 2.3. Cytosolic Ribosomal Proteome Reprogramming

Protein abundances are not predictable from respective transcript abundances due to regulatory mechanisms that remain elusive [60]. For instance, features of mRNA affect initiation, elongation, and termination of translation and thereby protein abundance. Inversely, mRNA degradation or sequestration [61,62] and protein degradation rates influence protein levels in a predictable manner [63]. Thus, in order to account for the limitations of predicting RP abundance based on transcript analyses, we performed shotgun proteomics after trypsin digest of ribosome complexes (Appendix A) enriched from whole root tissue extracts. We chose the same vegetative developmental stage and hydroponic cultivation conditions as in our transcriptome study. According to our transcriptomic results, ribosomes would more probably carry triggered RP structural changes 7 days after cold acclimation, thus we focused our proteomic analyses at this time point. We compared all ribosomal populations among them, taking samples harvested prior to the cold shift as control. Considering that stress-responsive RP paralogs with functional roles should differentially accumulate in translating and non-translating ribosome complexes, we investigated differential accumulation in polysomic compared to non-translating fractions and then focused only on the translating fractions. We omitted the 80S fraction because it can be a mixture of translating and non-translating mRNA-free complex species [38]. Moreover, we used three independent genotypes, wild type Col-0 and two independent knock out mutants of the Arabidopsis REIL proteins (*reil*-*dko*), i.e., cytosolic RBFs involved in late 60S maturation [34,39]. This allowed us to shed light on cold-triggered RP changes that are likely to occur in the early nucleolar biogenesis, since in the absence of REIL, 60S subunits are unable to mature during cold [38]. Thus, by including the *reil-dko* mutants we obtained cold response changes happening independent of REIL function that might trigger the well documented REIL-factors necessity during cold.

### 2.4. Substoichiometry in Non-Translating Versus Translating Ribosome Complexes

We tested whether RPs ubiquitously present in all samples (i.e., 66 RPs outlined in Appendix A) were significantly substoichiometric during cold. We found that upon acclimation some RPs significantly change their relative abundances in non-translational LSUs compared to polysomes (Figure 4). The changes are presented as the natural logarithm of 60S to polysome ratios, which means that a positive (Figure 4—purple) value represents a protein that is more abundant in the 60S fraction. Conversely, a negative (Figure 4—yellow) value means that a protein is more abundant in polysomes. There are several groups of responses in the context of RP stoichiometry. For example, specific paralogs from the RP families uL16, uL15, eL32, eL14, eL36, and eL18 are more abundant in the 60S fraction at 20 °C in Col-0 accession (i.e., the putative control or canonical RP composition for the WT genotype alone). The mentioned RPs are depleted from polysomes in relation to the non-translating 60S at 20 °C in the WT genotype. In other words, these RPs would be considered canonically more abundant in the non-translating fraction of Col-0 accession. After a shift to 10 °C, if these proteins were no longer more abundant in the 60S subunit and the ratios would be nearer to zero; these RPs would be considered substoichiometric in comparison to the previous canonical composition, which would entail that during cold these RPs are more abundant in the polysomes compared to their previous state. 

The canonical stoichiometry shared by the three Arabidopsis genotypes at optimized temperature features uL15 paralogs aB and aC as intrinsically more abundant in the 60S subunit as compared to the polysomes. Similarly, paralog C from family eL18 is more abundant in the 60S subunit as compared to the polysomes. These increased abundances imply that fewer protein units from the mentioned RPs are associated with actively translating polysomes at optimized physiological conditions. Oppositely, the canonical stoichiometry between 60S and polysomes is lost during cold acclimation. At 10 °C, plants exhibit relatively decreased abundances in polysomes as compared to 60S of paralogs A, C, and D from families uL3, eL28, and eL13, respectively. Interestingly, the canonical stoichiometry at 20 °C relates to a surplus of proteins in the 60S, which is then lost during cold acclimation, where the stoichiometry is subtractional, i.e., there is a deficit of proteins in the free 60S subunit pools. A list of the paralogs of RP families with altered stoichiometry, including trends that are genotype-specific, can be seen in Table 1.

Genotype-specific trends: We found indications of potential paralog exchanges or opposite 60S to polysome fold changes. For instance, during cold, the B or C paralogs potentially replace uL6_RPL9D. Similarly, P1.P2_RPP1A, canonically less abundant in 60S, is potentially replaced by B and C paralogs that became more abundant in the 60S fraction during cold. eL32_RPL32A is canonically more abundant in the 60S and less abundant during cold acclimation, while the B paralog is less abundant at 10 °C and already at 20 °C in the *dkos*. P1.P2_RPP2B, uL10_RPP0A or B and P1.P2_RPP1C are more abundant in the 60S LSU during cold acclimation in the WT, while already more abundant at 20 °C in the *dkos*.

### 2.5. Cold-Induced Changes in Active Translating Polysomes

In order to identify candidates that most likely represent functional remodeling in an acclimation context, we delimited our search focusing in changed low-oligomeric polysomal complexes (Figure 5). Polysomes are mRNAs loaded with several 80S translating monosomes and can be taken as a proxy of the translational fraction of ribosomes in the cell. Furthermore, grabbing entire roots or shoots for proteomics entails diluting *de novo* synthesized potentially specialized ribosomes with the pre-existing species. Therefore, candidate proteins that are present/absent during cold may be highly significant in a biological context. We weighed equally candidate proteins that are present or absent in a specific temperature-genotype combination (i.e., proteins standing in the red dashed lines in Appendix A) and those amenable to mean comparison between conditions. Relative quantitation and visualization of the changes through a bi-plot enabled us to encounter shared and significant responses among genotypes (Figure 5).

Shared significant cold responses common to *dko1*, *dko2,* and WT in Arabidopsis roots are clear in polysomal complexes. RPs uL30_RPL7C, eS24_RPS24A, and eL20_RPL18C are significantly less abundant in polysomes after seven days of shift to 10 °C, whereas eL14_RPL14B, eL34_RPL34A, eL42_RPL36aB or aA, and eL39_RPL39A or C are significantly more abundant in polysomes upon cold acclimation. Furthermore, genotype-specific trends can be inferred by analyzing each genotype and dataset individually (Appendix A). The trends coincide with the highest increases and lowest decreases in RPs according to fold change depicted in Figure 5**.** We picked robust changes based on their repeatability between datasets. Some changes appeared further robust taking into account that several RPs with unique peptide identifications from the same family follow the trends, e.g., uL30 paralogs in Appendix A are depleted during cold from all the genotypes.

WT-specific responses: Reprogramming of ribosomes in WT Col-0 entails decreased abundances of at least P1.P2_RPP1C and possibly P1.P2_RPP1A and 2A. These are components of the P-stalk. Abundances uL16_AtRPL10 increase during cold. The cold responsiveness of P-Stalk elements is not visible in two independent *reil*-*dko* mutants.

*dko*-specific responses: The *dko2* appears to be more responsive in terms of the number of log fold-changed RPs in the polysomes. eL34_RPL34B is enriched during cold and P1.P2_RPP0A or B are depleted during cold. Both changes are consistent with their relative abundance in the 60S fraction (Figure 4).

Candidates derived from presence/absence calls: Most of the candidates that appear to be present or absent during cold appear to be spatially related to the PET or P-stalk region, i.e., eL39_AtRPL39, uL10_AtRPPO, eL22_AtRPL22. eL22_AtRPL22B was amenable to relative quantitation in the first dataset and it is enriched during cold in both genotypes, in the second dataset this RP is also cold responsive in the WT and is only present at 20 °C in the *dko2*.

### 2.6. Spatially Constrained Cold-Triggered Ribosome Heterogeneity

After mapping significant RP transcript or protein changes into a 3D rendering of the plant cytosolic ribosome, we observed the recurrent spatial closeness in cold responses. Thus, we used the methodology detailed in the GitHub repository COSNet_i_ (https://github.com/MSeidelFed/COSNet_i, accessed on 30 April 2021) to formulate a statistical testing scheme that enabled us to probe for spatial enrichment in the ribosome complex, using both an available Cryo-EM structure and our own omics data. The test selects, through a random walk, coherent ribosomal regions based on protein–protein interactions at a given distance threshold (i.e., we assume structural proximity in interpreted Cryo-EM densities as a proxy for biochemical interaction) and test whether the proportion of significantly changed RPs in these regions differs to that of the entire ribosome. The null hypothesis is that significant changes are randomly scattered throughout the ribosome structure. Thus, the method uses the Fisher exact test to detect significant differences in the proportion of up, down, or inversely regulated paralogs (Appendix A). Using this method, we tested our transcriptome and proteome datasets to verify if specific regions are modulated during early ribosome biogenesis in plants undergoing cold acclimation (Figure 6).

We tested three different levels in both the transcriptome (xT) and proteome (xP) data (Appendix A): (1T) inversely regulated, (2T) inverse and upregulated, (3T) inverse, down, and upregulated as detailed in Figure 3. (1P) 60S to polysome ratios, (2P) polysome or (3P) both instances as detailed in Figure 4 and Figure 5. At a significance level of Q value < 0.05 (i.e., Bonferroni adjusted *p* value), the spatial regions identified as significantly changed upon cold acclimation are depicted in Figure 6. At the transcript level, the molecular species from the 60S—P-Stalk and the 40S–60S transition of the ribosomal head were significantly enriched in relevant cold-induced changes. At the proteome level, abundances of RPs that belong to the polypeptide exit tunnel (PET) region were significantly changed during cold; this region includes RPs inserted into the tunnel. Moreover, since the Bonferroni correction can be statistically stringent, we report several regions that resulted significant with *p* values < 0.05 (Table 2 and Appendix A).

Even though not all significant changes remained below 0.05 after the multiple test correction, the same regions emerge across datasets and testing schemes. For example, the P-Stalk is a major reprogramming event at the transcriptome level as well as the PET at the proteome level. Both are supported by multiple cases of significance across our results. Similarly, the region containing the ribosomal head at the 60S–40S transition zone is significantly changed at the transcriptome level. Finally, several regions are outstanding due to their recurrence, namely the uL30-uL3 containing regions, which exhibit *p* values < 0.05 both at the transcriptome and proteome level.

### 2.7. Paralog Specific Cold Responses—uL30 Family

The four uL30 paralogs were consistently changed during cold acclimation both at the transcript and protein level. The C and D paralogs were inversely regulated 7 days after cold acclimation at the transcript level (Figure 3). At the protein level, The C paralog became more abundant in the 60S subunits as compared to polysomes (Figure 4) while being significantly less abundant in the polysomal fraction (Figure 5). Paralogs B, C, and D exhibited a two-fold decrease during acclimation in WT and *reil*-*dko* mutants polysomes (Appendix A) while the A paralog was only detected in polysomes in the shoots during cold (Appendix A). Regions containing uL30 RP were frequently enriched for cold-relevant changes (Appendix A) and remained significant at the proteome level after Bonferroni correction (Figure 6). Thus, we investigated the length to which uL30 functions are known in model eukaryotes, aiming to determine if there is a relation between uL30s, conserved cold responses, and the spatial constraints encountered. We interpreted reported functional aspects of this gene family in other eukaryotes using a phylogenetic tree alignment of the protein coding regions of *Arabidopsis thaliana*, *Saccharomyces cerevisiae,* and *Homo sapiens* (Figure 7).

Yeast—*Saccharomyces cerevisiae* (Sc)—has two paralog genes (i.e., ScRPL7A and ScRPL7B) coding the uL30 homologs. According to a phylogenetic tree alignment of the protein coding regions (Figure 7), yeast homologs are the nearest to a “common” eukaryotic ancestor as compared to plants and mammals. Two paralogs (i.e., HsRPL7 and HsRPL7A) and one alike-protein (i.e., HsRPL7L1) code the *Homo sapiens* (Hs) homologs, which differ in their degree of divergence as compared to a common hypothetical ancestor. Still, the three proteins cluster together, suggesting low diversification between them. In Arabidopsis, AtRPL7A and D are the least diversified in comparison to the common ancestor. AtRPL7A is adjacent to Sc homologs in terms of nucleotide substitutions, whereas AtRPL7B and C appear to have diversified substantially and are closer to Hs homologs. Notably, At paralogs do not cluster together as do Hs and Sc, suggesting that these might have diverged the most.

## 3. Discussion

### 3.1. Different Types of RAP Transcripts Mediate the Initial and Long-Term Responses to Temperature Acclimation

The transcriptomic fingerprints of cold acclimation in *Arabidopsis thaliana* are compiled in Genevestigator [64], where experiments across mutant and wild-type genotypes [65,66,67,68,69,70,71,72,73] make clear that a major transcriptome reprogramming occurs [74]. During cold, changes in gene expression remodel machinery and metabolic pathways to enable acclimation. There are two main groups of transcriptional responses [37]: immediate responding genes (responding after 1 h) that modulate the direct response, and late responsive genes (responding after 1 day) that keep changes more permanently, thus enabling acclimation. Between these two classes of early responsive genes, translation-related genes are a main hub mediating cold acclimation [32]. The transcriptome of ribosome-associated protein (RAP) transcripts responds at 1–7 days of cold acclimation [34] and there are paralog-specific abundance shifts (Figure 2 and Figure 3 in this manuscript and Figure 3 and Figure 4 from Martinez-Seidel et al., 2020) indicating potential paralog functional divergence [7] adjusted to different temperatures.

In order to extend previous insights, in this manuscript we interrogated the cold response dynamics of transiently binding RAPs and structural RPs. A larger amount of transiently binding RAPs (including RBFs and TIFs) become differentially regulated after one day of temperature acclimation (FDR < 0.05 = 482, Appendix A) as compared to 7 days (FDR < 0.05 = 236, Appendix A). In consequence, most transiently binding RAPs can be classified in the second group of late responsive genes proposed by Seki et al. (2002). Next, we inquired about ribosome biogenesis factors (RBFs) as effectors of the transcriptional response. Genes coding for RBFs and putative nucleic acid binding proteins are specially regulated 1 day after shifting to cold. Apart from one transmembrane chaperone, all the other genes from 24 significantly changed RBFs are related to nucleic acid binding and rRNA transcription and processing (Appendix A) [2,75,76,77]. At the later time-point, i.e., day 7, only three RBFs were differentially regulated, two of them featuring nucleic acid binding domains and rRNA processing functions. On the other hand, continuously regulated RBF genes in both 1 and 7 days are mostly related to rRNA processing pathways. Cold-triggered modulation of rRNA processing and decrease of pre-rRNA species has been reported in rice, maize, and Arabidopsis [43,78,79]. Rather than just rRNA synthesis arrest, we argue that in essence, cold exposure seems to reprogram the synthesis of new ribosomes by modifying rRNA via transcription and processing, in order to rearrange the concomitant and serial assembly of new cold-related protein paralogs. Accordingly, co- and post-transcriptional modifications on the rRNA topography may lead to different possibilities of protein interactions and stoichiometry [80].

Assembly of RPs during biogenesis relies on an orchestrated serial “entry and exit” of sets of RAPs. For instance, early in the nucleolus, the 60S GTPase-Associated Center (GAC), Polypeptide Exit Tunnel (PET), and Peptidyl Transferase Center (PTC) start to be formed and mounted with some of their RPs. In our datasets, structural RPs show the opposite trend as compared to total RAPs, with a larger number of RPs being differentially regulated after 7 days of acclimation (Figure 3 and Appendix A). Thus, structural RPs might constitute a third group of cold responsive genes regulated in response to altered ribosome biogenesis in order to store a molecular memory of low temperatures as structural changes in the ribosome.

### 3.2. Cold-Triggered Reprogramming Indicates That Spatial Constraints Adjust the Ribosomal Proteome

The initial transcriptome regulation of RBFs and other RAPs at day 1 of acclimation is followed by the differential regulation of the transcriptome coding for structural RPs at day 7. Thus, seven days after the initial cold cue is the essential time-point to understand what the initial transcriptome regulation implies for the ribosome structure. Previously we have shown that plants alter their RP stoichiometry upon a transition to cold [36,38]. Here, we outline that paralog-specific cold changes deviate from a canonical stoichiometry. The cold RP substoichiometry is related to molecular species from the PET region, whereas transcriptome RP substoichiometry is related to molecular species from the P- Stalk and the ribosomal head. Within the significantly enriched regions, paralog-specific exchanges indicate that functional divergence among RP paralogs occurred. Redundant paralogs are likely to sub and/or neofunctionalize [6]. In our case, we present changes that include simultaneous down and up-regulation of RP paralogs within the same family, or up- down- regulation of a single paralog within a RP family, suggesting paralog adjustment to temperature. Moreover, spatially adjacent changes suggest that evolutionary forces driving paralog divergence could be structurally constrained, either by co-evolution of binding sites for RPs or RP paralogs [81], or fundamentally influenced by the concerted mounting of RPs that happens during ribosome biogenesis [82]. Plant ribosome biogenesis, for example, features several specific factors that mediate processing steps of the 60S LSU [1,2]. The RBFs assisting and mounting RPs are not exclusive but rather assist with the mounting of several RPs at a specific stage. Therefore, a single RBF could affect a group of similarly located RPs. Consequently, the regional adjustments in the ribosomal proteome can be expected and may be used to further modulate cold acclimation by enabling ribosomes to select transcripts for translation.

Selecting ribosomal regions and studying the interactions between all ribosomal components was initially enhanced by the first high-resolution structure of an eukaryotic ribosome that helped to decipher in detail all the interactions [83]. This type of information allowed us to define coherent ribosomal regions, which is the most influential step towards finding spatial enrichments. Several considerations and criteria needed to be met. For instance, ribosomal regions of biological interest could be defined based on rRNA domains and accessory proteins [84]. However, pre-knowledge biases must be avoided while selecting coherent independent regions in order to fulfill the statistical test assumptions. We achieved this by weighting protein–protein interaction networks with the percentage of contact coverage between proteins. This allowed us to draw spatial constraints by defining a transit probability during the subsetting procedure. We selected an 8 Ångströms threshold based on the quality of the resulting regions (e.g., connectivity and biological accuracy) and literature consensus on distances between amino acid residues within a protein structure [85]. Later on, we used the coherent regions to test if the proportion of significances in each RP subset compared to the RP universe changes. The statistical test selected was the Fisher exact test [86,87], followed by the strict Bonferroni correction [88].

Spatial adjustments of the ribosomal proteome may be at the core of ribosome specialization. The functional centers of ribosomes actively communicate to perform ribosomal functions. The communication among functional centers is optimized by the coevolution of a non-random RP network [89]. These RP graphs harbor specific neuron-like properties that led to the realization of RPs being instrumental in the process of information transfer across ribosomal complexes [90,91]. For example, mutating specific amino acid residues from uL5 in yeast leads to impaired inter-subunit communication, which in turn causes structural alterations in 40S and 60S rRNA [92], highlighting the large scale of information flow within the network. Similarly, uL3 in yeast plays a role in the coordination of the elongation cycle by communicating the tRNA site status to the elongation factor binding region and the peptidyltransferase center [93]. Thus, small changes in RPs can lead to vast rearrangements in the RP network and adjustment of ribosomal function. We used the plant RP network to define coherent ribosomal regions and used those regions to interpret our omics data in their structural context. We found that altered stoichiometry in specific RP and RP paralogs is correlated to modulated ribosomal substructures during cold acclimation in plants, suggesting for example that uL30 acts as a communication hub that restructures the ribosome during cold. Moreover, we argue that after stopping ribosome biogenesis during cold, the modulation of these substructures is likely to occur in the early nucleolar biogenesis as a response to maintain protein synthesis during acclimation.

### 3.3. Cold Ribosomal Protein Changes during Early Biogenesis

The ribosome physiology of cold acclimation indicates that Arabidopsis wild type plants start accumulating 60S free subunits rapidly after a cold cue and steadily during the first 7 days of acclimation [38]. By comparison, *reil-dko* genotypes present an accumulation of 60S subunits before the shift to cold, which are rapidly depleted upon a cold cue and are only slowly replenished across the acclimation period until a wild type-like abundance level [38]. Nevertheless in spite of the accumulation of 60S subunits in the mutants, these complexes appear not to be competent since *reil* mutants fail to restart growth after 7 days of acclimation [34]. Thus, the 60S subunits that are accumulated are most likely defective. Here we show that such defects are probably stemming from nucleolar biogenesis, since beyond wild type-specific or *reil-dko*-specific RP changes, there is a spatially constrained cold substoichiometry supported by several RP changes shared among genotypes.

Deviations from the canonical RP composition comprise at the proteome level the PET region. Specific RP changes entail at least increased abundances of uL15_RPL27aB, uL15_RPL27aC, eL18_RPL18C in the LSU as compared to polysomes at 20 °C and increased abundances of uL3_RPL3A, eL28_RPL28C, eL13_RPL13D in polysomes as compared to LSUs at 10 °C. Importantly, the intrinsic increased abundances across several RPs in the 60S population before the temperature shift indicates a surplus of proteins bound to the non-translational 60S subunits. RPs can exhibit promiscuous binding in archaeal ribosomes with some proteins being present at more than one location per 50S ribosome [94]. Moreover, metazoan cytosolic ribosomes acquired novel expansion segments (ES) logarithmically over the past two billion years as compared to archaeal and bacterial ribosomes [95,96]. Metazoan RPs have also diversified, increased in number, and in parts diverged [97,98,99,100,101]. Thus, it is likely that the availability of novel rRNA expansion segments and the diversification of RPs enhances promiscuous binding of multiple RP or RP paralog copies per ribosomal particle in metazoans and specially also in plant ribosomes. Moreover, storing these “RP abundant” complexes in a non-translational fraction might be a on the run strategy to rapidly tune the ribosomal network and meet translational needs. Consequently, the surplus of RPs in 60S subunits supports one of our previous notions, i.e., “that Arabidopsis may buffer fluctuating translation by pre-existing non-translating ribosomes before *de novo* synthesis meets temperature-induced demands” [38]. Once a shift to cold occurs, the stoichiometry of 60S subunits is intrinsically subtractional [102], indicating that the surplus population that was stored to meet translational demands likely got depleted. On the other hand, in active translating polysomes, the RP stoichiometry changes bilaterally, that is, abundances of bound uL30_RPL7C, eS24_RPS24A, and eL20_RPL18C significantly decrease during cold acclimation, while the abundances of bound eL14_RPL14B, eL34_RPL34A, eL42_RPL36aB or aA, and eL39_RPL39A or C significantly increase. This indicates that active ribosomes need a finer adjustment of their structural proteome to meet translational demands during cold.

### 3.4. Cold Dynamics of Ribosomal Protein Assembly

Assembly and specific interactions of different RP or RP paralogs may be a product of the effect of lower temperatures on protein stability and folding states. Agozzino and Dill [103] described a model for the sequence adaptation of proteins according to temperature changes and argued that the least stable proteins are the ones that adapt faster to temperature changes and help the organism adapt faster. In this process, chaperones and their properties are essential as capacitors of cellular evolution. One process that can alter folding states of globular proteins is cold denaturation [104,105,106,107]. Besides heat, cold can denature proteins due to the disturbance of the forms of noncovalent bonding that are responsible for the folding state at physiological conditions. Thus, cold denaturation is essentially dependent of the protein structure and the hydrophobic effect [106]. Due to different amino acid composition and structural features, the temperature in which cold triggers denaturation can vary for different proteins.

At 10 °C in our experiments, cold denaturation could account for the spatially changed PET region in both transcript and proteome data (Figure 6 and Table 2) and the choice of assembling different RP or RP paralogs. Many RNA-binding proteins have linkers as intrinsically disordered regions (IDRs) in their natural native state, which confer flexibility and plasticity to the RNA-binding domains [108]. In RPs, IDRs are disordered extensions that can stretch to different parts of the subunits. Extensions can penetrate the core of subunits and transition to a more ordered structure and assist rRNA folding, and different categories of extensions may have distinct functions in assembly stages [109,110,111]. Moreover, many eukaryotic RPs’ disordered extensions interact with the rRNA expansion segments (ES) around the peptide exit tunnel (PET) [83], and from the P-stalk to the L1 side. ES are present in the surface of both subunits, but in the 60S they are more abundant and form a nearly continuous ring around the PET. It may be that cold structurally modulated RP paralog extensions can lead to different interactions with rRNA helices and ES due to different folding states, leading to heterogeneous subpopulations of subunits and ultimately ribosomal complexes.

### 3.5. Cold-Induced Ribosomal Protein Substoichiometry Co-Localizes With Rei1 Binding Site

In yeast, Rei1 inserts its C-Terminus inside the PET in order to probe the integrity of the tunnel [41]. Rei1 binds in the vicinity of the PET exit [112]. The binding site was initially thought to be eL24, but the deletion of eL24 did not prevent the factor to bind to the LSU [113]. Cryo-EM structures [41] show Rei1 to be bound to eL22 in its tunnel inserted state (see Appendix A). Residues 355–385 might enhance the opening of the PET to achieve C-Terminus insertion [41]. In plants, Rei1-like (REIL) proteins mediate low temperature perception and the molecular implications of REIL loss compared to yeast homologous mutants are similar [34]. Thus, altered RP stoichiometry near or inside the tunnel can be very relevant for REIL proteins since it would co-localize with their functional and binding site.

The ribosomal proteome surrounding the PET is significantly remodeled during cold. These rearrangements may be the cause of REIL necessity because they occur indistinctly in wild type and two independent *reil*-*dko* genotypes. Thus, rearrangements directly associated or happening after the 60S-REIL interaction are omitted. The encountered remodeling implies altered paralog dynamics around REIL’s binding and operating site. Hence, among possibilities, the tunnel structure could be weaker or binding sites for canonical maturation factors may be lost, and then, the appearance of a specialized protein to prove the integrity of the rebuilt tunnel, such as REIL, would be ideal. As an example of paralog-specific changes, within RP family eL22 (blue in Appendix A), paralog eL22-RPL22B is decreased in the WT-60S as compared to the polysomes during cold, potentially signaling a cold-specific binding event for REIL. The eL22 paralogs have effectively subfunctionalized in Drosophila [114]. Additionally, in yeast, paralog-specific phenotypes of eL22 are related to translational control of the serine and methionine metabolic pathways [115]. Thus, variations in eL22 paralogs can effectively produce functional divergence and may be used by plants to enhance the REIL-60S interaction.

### 3.6. Cold Dynamics of uL30 Paralogs Could Orchestrate Spatially Constrained Rearrangements in Ribosomes

Paralog rearrangements can characterize ribosome specialization in eukaryotes [13,116,117]. Similarly, as we have shown, several paralog-specific changes characterize cold-acclimated ribosomes in Arabidopsis. We interrogated the uL30 gene family as an exemplary case of induced changes that may occur as a response to cold early on during nucleolar biogenesis. This RP family is also responsive to cold in other higher eukaryotes, for example, in wood frog *Rana sylvatica* RsRPL7 is upregulated during cold acclimation, conferring resistance to freeze tolerance [118]. Functionally, uL30 mediates the pre-RNA cleavage at site C2 in ITS2, separating precursors of rRNA into 5.8S and 25S in yeast [119]. In Arabidopsis, uL30 is encoded by four paralog genes, two of them, *rpl7a* and *b*, have a leaf phenotype [120], but the molecular implications of the loss have not been evaluated. An important observation that explains a potential paralog-exchange was reported for the *Homo Sapiens* RPL7 (i.e., close homolog of AtRPL7B and C as detailed in Figure 7), which inhibits translation of specific mRNAs including its own [121,122]. Our work shows that the transcript of AtRPL7C becomes upregulated during cold acclimation, while its proteoform becomes depleted in the translational fraction during cold. This implies a translational regulatory mechanism that relies on the increased transcript abundance of the C paralog. Interestingly, the C and D and not the A and B paralogs have been reported to contain upstream open reading frame (uORF) sequences that most likely regulate translation [123], providing a hypothetic testable mechanism. In shoots, AtRPL7A (i.e., the closest homolog to ScRPL7A and B) is cold-specific according to presence/absence calls (see Appendix A). 

Evidence from model eukaryotes suggest that paralog exchanges in the uL30-AtRPL7 family may trigger regionally constrained ribosome heterogeneity. More specifically, knocking out yeast uL30 homologs triggers defects in the pre-RNA assembly step 27SA3 to 27SB, causing in the end reduction in four RPs that surround the polypeptide exit tunnel [124], similar to what we report. Therefore, decreased uL30s from translationally competent complexes could orchestrate structural rearrangements of the PET. From a ribosome perspective, remodeling the PET would be one of the most efficient mechanisms to constrain translation of specific mRNAs by stalling nascent peptides. The PET is one of the major determinants of ribosome velocity through a transcript [125] mainly due to the amount of negative charges across the tunnel slowing down positively charged amino acids; hence, the PET architecture would be highly susceptible to slight modifications that would make it improbable for specific positively charged peptides to be translated without stalling. Thus, remodeling the tunnel might enable ribosomes to rapidly shape the cold proteome by translating specific transcripts.

### 3.7. REIL Concomitant Ribosome Reprogramming Argues Potential Specialization

REIL proteins in plants have acquired functional changes including putative novel roles in processes such as cold acclimation [34]. Within the possibilities of acquired functions, and aligned with evidence, REIL factors either directly or indirectly enhance regional restructuring around the P-stalk during cold acclimation (Figure 5). In yeast, Rei1 interaction is followed by changes in the assembly of the last ribosomal component, namely the P-Stalk [45]. The P-Stalk recruits proteins from the large GTPase family directly assisting translation initiation [126], because this family contains the translation initiation factors (TIFs) [127]. Thus, remodeling the P-Stalk is a tremendous opportunity to achieve selective translation by directing TIFs towards specific mRNAs. At the transcriptome level, the P-Stalk and the ribosomal head are enriched in cold significant changes. At the protein level, the observed changes in several members of the P-stalk in polysomes are consistently wild-type specific (Appendix A), arguing that the lack of REIL may impair the P-Stalk remodeling that is necessary for successful acclimation. Functionally, both regions are involved in mRNA translation, the P-Stalk recruiting TIFs and the head laying right next to the mRNA passage sites on the ribosome. Therefore, a transcript-mediated mechanism used by the ribosome to constrain total or specific translation events could be initiated by these ribosomal components. Similarly, at the proteome level, the PET is significantly enriched in RP cold-induced changes. The PET is the transit point for every single polypeptide that is synthesized. Also described as the “birth canal of biology” [95], it is as such also a hotspot to constrain total or specific translation for ribosomes according to external stimuli. These reprogramming events could represent fast strategies used by the ribosome to constrain translation upon cold acclimation, mediated partly by REIL newly acquired functions.

## 4. Materials and Methods

### 4.1. Plant Material

The Arabidopsis double homozygous *reil1-1 reil2-1,* and *reil1-1 reil2-2* mutant lines (*dko1* and *dko2*) were created by crossing the T-DNA insertion mutant SALK_090487 (*reil1-1*) with GK_166C10 (*reil2-1*), i.e., double knock-out mutant 1 (*dko1*) or SALK_040068 (*reil2-2*) [39], i.e., double knock-out mutant 2 (*dko2*). The Nottingham Arabidopsis Stock Centre made the Salk lines available (http://signal.salk.edu/, accessed on 30 April 2021). The GK line was obtained through the GABI-Kat program [128]. Homozygous lines were verified by PCR amplification of genomic DNA [39]. The genetic background of the initial mutants is accession Col-0. Hence, all experiments were in comparison to this wild type and *Arabidopsis thaliana* Col-0 was used as positive control throughout this study.

### 4.2. Growth Conditions

Plants were grown in a hydroponic system. Autoclaved assembled glassware with metallic meshes were used to place Arabidopsis seedlings at vegetative developmental stage 1.0 [55] for sterile culture on top of liquid medium (Appendix A). Arabidopsis seedlings were germinated and transferred onto the mesh using small blocks of solid agar. Transferred plants were grown within the translucent glass pots with non-airtight glass lids until stage 1.10 inside growth chambers with a 16 h/8 h day/night cycle. Plant age at stage 1.10 was approximately 21 days after sowing. The hydroponic solution was Murashige and Skoog medium [129] supplemented with 2% (*w*/*v*) of sucrose. The growth chamber environment was kept as reproducible and controlled as possible. The chambers were set to a light intensity of 40 µE. The effective light intensity within the glassware was ~30 µE. The temperature inside the glassware was externally controlled but temperature inside the glass pots increased during illumination. External 10 °C of the growth chamber generated ~12 °C inside the glassware, external 20 °C and 30 °C, ~23 °C and ~31 °C during the light phase, respectively. Plants were temperature-shifted in the middle of the light phase. Minor fluctuations of light intensity and temperature were encountered according to the position within the growth chamber. These fluctuations were accounted for by rotating the position of glassware daily. Block and tray effects were minimized by pooling material across jars and different trays when collecting biological replicates. 

### 4.3. Dry Weight Measurements

A BP210S balance (Sartorious AG, Göttingen, Germany) was used for weighing. To improve balance precision at different locations, the balance was temperature-equilibrated to locations at least a day prior to measurements (Appendix A). Accuracy of weight determination after re-location of the balance was controlled by small metal calibration weights (Appendix A). Empty bags were folded to fit the balance and to speed up weight equilibration. The dry matter content of hydroponic plant material was determined using the weight loss after drying. Whole plants were taken from hydroponic cultivation and placed inside wax-layered and pre-weighed bags after 48, 72, 84, 144, and 168 h after shifting to either 10 or 30 °C, or continuous growth at 20 °C. To eliminate excess liquid from root material, roots were rapidly pre-dried with filter paper three times within a total period of 2–6 s. Adherent solid agar pieces were removed thoroughly without compromising the speed of weighing. The initial weights of empty and full bags were registered to calculate sample fresh weight. Full bags were dried at 70 °C for three days. Subsequently, the bags were temperature-equilibrated 1 h inside a desiccator and the dry weight of the full bag and emptied dried bag determined to calculate sample dry weight. Weight loss of bags was corrected by measurement of initial and dried weights of 155 empty bags (Appendix A).

### 4.4. Microarray-Based Transcriptome Analysis

Total RNA from hydroponic Arabidopsis root samples with fresh weights of 25.1–26.8 mg were harvested at time points 0 h, i.e., at 20 °C before shift to cold, and 1 or 7 days after shift to 10 °C. RNA was isolated using the RNeasy Plant Mini Kit (QIAGEN, Hilden, Germany) according to the manufacturer’s instructions [130] with minor modifications. Briefly, 110 µL of RTL lysis buffer was used to extract RNA. 30 µL of RNase-free water was added to elute the RNA. Three replicates were prepared per time point (*n* = 3). The RNA integrity number (RIN) was determined by an Agilent 2100 Bioanalyzer and 2100 Expert software (Agilent Technologies, Santa Clara, CA, USA). RNA samples with RIN of more than eight were used for 4x44K Agilent expression profiling (ATLAS Biolabs, Berlin, Germany). The expression data sets were uploaded to the Gene Expression Omnibus (https://www.ncbi.nlm.nih.gov/geo/, accessed on 30 April 2021) and are available through accession number GSE144916. 

### 4.5. Transcriptome Data Analyses

A matrix of one-color-processed green signal values (gProcessedSignals) was obtained from ATLAS Biolabs (Berlin, Germany). Company signal processing comprised multiplicatively de-trended background subtraction, correction for spatial effects, and removal of outlier pixels. A subset of the complete transcriptome matrix was created that contained only probes encoding open reading frames of cytosolic RPs (based on Appendix A from Martinez-Seidel et al., 2020) [131] or total RAPs [34]. The resulting matrix contained in part redundant probes of gene models. The respective signals of redundant probes were analyzed separately to internally control for the quality of differential hybridization and to recognize potentially differential hybridization of included splice variants. The resulting Ln subset matrix was quantile normalized using the normalize.quantile function of the preprocessCore package (https://github.com/bmbolstad/preprocessCore, accessed on 30 April 2021) version 1.46.0. for R statistical computing and graphical visualization language (https://www.r-project.org, accessed on 30 April 2021). Batch effects that result from between chip variation of microarray experiments were accounted for using comBAT [132], an empirical Bayes normalization algorithm. The distribution of the resulting normalized probe hybridization abundances was tested using a dependency of the R-package fitdistrplus [133] through R functions located in the GitHub repository RandodiStats (https://github.com/MSeidelFed/RandodiStats, accessed on 30 April 2021). The distribution shapes were visualized by Cullen and Frey graphs featuring the kurtosis against the square of skewness (Appendix A) and indicated an approximately Gaussian distribution. Thus enabling statistical testing by fitting of linear models using the R-package limma [134]. A generalized linear model (GLM) procedure without parametrization of the mean and variance gave equivalent significance (*p* values) (Appendix A). The resulting *p* values were corrected for multiple testing using the false discovery rate proposed by Benjamini and Hochberg (BH-95) [135]. K-means clustering [136] was applied to the autoscaled matrix of differentially expressed genes of cytosolic ribosome proteins and expression trends of cold acclimation were identified. Based on these trends, the RP families were tested for paralog-specific differential gene expression. We considered three possible paralog-scenarios. (1) The entire gene family or at least one paralog within has increased gene expression, (2) the entire family or at least one paralog has decreased gene expression, or (3) at least two paralogs of a gene family are inversely regulated during cold acclimation.

### 4.6. Cytosolic Ribosomal Proteome Preparation

We obtained non-translating and translating ribosomal complexes using previously reported methods [49]. Due to the complexity of ribosome preparation and respective ribo-proteome analysis, we performed two experiments resulting in dataset 1 (DS1) and dataset 2 (DS2) respectively. Each dataset contained a pooled Arabidopsis Col-0 sample and either a pooled double mutant sample of *reil1-1 reil2-2* (*ds1*) or *reil1-1 reil2-1* (*ds2*) for a total of 4 biological replicates used for the pooled genotype statistical analyses. Samples prior to shift were compared to samples prepared at 7 days after 10 °C cold shift. Polysome extraction buffer [137,138] was used to lyse membranes and isolate ribosome complexes from frozen and ground plant tissue. The extracted ribosome complexes were loaded onto sterile ultracentrifuge tubes, thinwall polyallomer tubes of 14 mL volume and 14 × 89 mm dimensions (Beckman Coulter, Brea, CA, USA). Sucrose gradients were prepared from 15%, 30%, 45%, and 60% (*w*/*v*) stock solutions. Ultracentrifugation was 14.5 h at 33,000 rpm using an Optima LM-80 XP ultracentrifuge and SW 41 Ti rotor (Beckman Coulter, Brea, CA, USA). The ribosome complexes separated according to their sedimentation coefficient [139] into 40S, 60S, 80S and low oligomeric polysomal complexes that were sampled into separate fractions monitoring rRNA absorbance at 254 nm wavelength. A programmable density gradient fractionation system was used (Teledyne Isco Inc., Lincoln, NE, USA). Resulting ribosome fractions were loaded onto regenerated cellulose membranes, Amicon Ultra-0.5 centrifugal filter units, with a 3-kDa molecular size cutoff (Merck, Kenilworth, NJ, USA). The filter units were washed iteratively with 500 µL of 0.04 M Tris-HCL buffer (pH 8.4) with 0.2 M KCl and 0.1 M MgCl_2_. Washing was repeated until the residual volume decreased to below 100 µL within 10 min of centrifugation at the recommended 5000–7000 rpm and 4 °C. Volume reduction by centrifugation was timed to terminate washing because each of the fractions contained different percentages of sucrose and hence required varying numbers of cleaning steps. Cleaned fractions were digested with trypsin by filter-aided sample preparation (FASP) according to previously reported methods [140,141].

### 4.7. Proteome Analysis by Liquid Chromatography—Tandem Mass Spectrometry (LC-MS/MS)

Tryptic peptides were loaded onto an ACQUITY UPLC M-Class system (Waters Corporation, Milford, MA, USA) hyphenated with a Q-Exactive HF high-resolution mass spectrometer (Thermo Fisher Scientific, Waltham, MA, USA). For dataset 1 (DS1), samples were separated by reverse-phase nano-liquid chromatography using a 125 min gradient ramped from 3% to 85% acetonitrile (ACN). Mass spectrometry was performed by a data dependent top-N tandem mass spectrometry method (dd-MS2) that fragmented the top 10 most intense ions per full scan. Full scans were acquired at a resolution of 120,000 with automatic gain control (AGC) target set to 3 × 10^6^, maximum injection time 100 ms, scan range 300 to 1600 *m*/*z* in profile mode. Each dd-MS2 scan was recorded in profile mode at a resolution of 15,000 with AGC target set to 1 × 10^5^, maximum injection time 150 ms, isolation window of 1.2 *m*/*z*, normalized collision energy 27 eV and dynamic exclusion of 30 s. Settings of dataset 2 (DS2) were slightly modified. We used a 132 min gradient ramped from 3% to 85% ACN. The dd-MS2 method fragmented the top 15 most intense ions per full scan. Full scans were acquired at a resolution of 60,000 with AGC target set to 1 × 10^6^, maximum injection time 75 ms, and a scan range of 300 to 1600 *m*/*z* in profile mode. The dd-MS2 scans were recorded in profile mode at a resolution of 30,000 with AGC target set to 1 × 10^5^, maximum injection time 150 ms, isolation window of 1.4 *m*/*z*, normalized collision energy 27 eV and dynamic exclusion of 30 s. The mass spectrometry proteomics data have been deposited to the ProteomeXchange Consortium via the PRIDE partner repository [142] with the dataset identifier PXD016292.

### 4.8. Proteome Data Analyses

The initial tandem LC-MS/MS chromatogram files of raw data format were processed with the MaxQuant software (Version 1.6.0.16). We analyzed label-free quantitative peptide abundances [143] (LFQ) across all samples. Peptides were annotated with *Arabidopsis thaliana* FASTA files obtained from the UniProt database [144] that contained 15,893 proteins reviewed in the Swiss-Prot subsection. All peptides, including unique and redundant peptides of structural cytosolic ribosome proteins were used for the subsequent analyses. Independent LFQ matrices were created for each of the two experiments, DS1 and DS2. Each matrix contained all ribosome fractions of an Arabidopsis Col-0 wild type to *reil1 reil2* double mutant comparison. Within the single fractions, the LFQ abundance of each small 40S subunit RP was normalized to the sum of abundances of all 40S SSU proteins. The LFQ abundance of each 60S subunit RP was normalized to the sum of abundances of all 60S LSU proteins per fraction, respectively. This normalization accounted for the varying amount of 40S SSU and 60S LSU subunits within each of the sucrose density fractions. After this normalization step, each cell in the resulting matrix (Appendix A) represented the fraction of each individual RP within its total subunit, either 40S or 60S defined by the following equation:NX_ij_ = X_ij_/ ∑[RPs]_i_
where ∑ [RPs]_i_ represents the sum of abundances of structural RPs belonging to the 40S or 60S subunit per sample. Finally, matrices belonging to independent experiments were merged yielding 149 potential structural RP paralogs or 123 RPs in total that were detectable in both experiments, DS1 and DS2. This matrix included non-unique peptides of paralog sets. As was observed with the transcript abundances of RPs, the distributions of protein abundances on average approximated normal distribution (Appendix A). Hence, GLM fitting with the Gaussian family of regression sufficed for statistical testing. We analyzed the fold changes between non-translational and polysomic fractions (Appendix A) or within the acclimated versus non-acclimated polysomic fraction (Appendix A) of each individual RP at day 7 after cold shift. The significance of substoichiometric RPs was tested using the three genotypes as joined validation for our statistics, namely, *dko1*, *dko2,* and Col-0. This decision provided robustness to our candidate selection and ensured that the significantly substoichiometric RPs are largely independent of REIL proteins.

### 4.9. Structural Analysis of Changes in Ribosome Protein or Transcript Abundance 

Since there is currently no complete, highly resolved *Arabidopsis thaliana* ribosomal structure that is publicly available, the reference structure used for this project was that of the translating *Triticum aestivum* (common wheat) cytosolic ribosome, published in 2010 [145]. PDB entry in the RCSB databank [146] is 4V7E. This structure was generated through modeling of known RP structures from bacterial and archaeal templates onto a *Triticum aestivum* Cryo-EM ribosome map. This model is canonically complete and contains 47 RPs within the 60S LSU and 33 RPs of the 40S SSU subunit. Four rRNA structures, one t-RNA bounded to the P-site, and an mRNA transcript are included. It is important to note that the 5.5 Å resolution holds true for the best-resolved regions of the reference ribosome structure but resolution may locally vary. The in part low resolution of the Cryo-EM map was manually confirmed by visualization of selected regions at the Electron Microscopy Data Bank (EMDB), e.g., accession EMD-1780.

The methodology is divided into four steps: (1) initial structural data preprocessing, (2) proximity network building, (3) structural region sampling and definition, and (4) statistical testing of enriched relative changes within structural regions, and has been compiled in the GitHub repository COSNet_i_ (https://github.com/MSeidelFed/COSNet_i, accessed on 30 April 2021). Code from this repository is written modularly, to reflect the steps of the methodology above, and is currently under development and testing to form a complete pipeline to encompass the entire workflow. To further explain the methodology: as preprocessing steps “Hetero atoms” (HETATMs) and duplicate atoms were removed from all proteins. Subsequently, the RP sequences were Blasted [147] to verify the correct annotation and renamed according to the last naming scheme [148]. Next, a proximity network was constructed of protein–protein interactions (based on structural proximity) omitting the rRNA, and based on a distance threshold of 8 Å [85] between individual amino acids coarse-grained to their center of gravity. Weights of this network’s edges were calculated as the proportion of inter-amino acid residue contacts between two proteins. A higher weight indicated a larger contact surface between two proteins. Networks were visualized using the R package *igraph* [149]. We sampled random regions from the proximity network defining a walk length and an iteration number. The weight of edges was used as transit probability (i.e., walking from x to y is written P_{x,y} = w_{x,y}/**w**_{x}, where **w**_{x} represents the sum of all weights from the outgoing edges of node x). A consensus walk from the iteration was calculated and pre-regions defined for every start node. Finally, we calculated the minimum set cover that spans the whole edge universe while minimizing the number of overlapping regions. The Fisher exact test allowed us to test if the selected regions were significantly enriched as compared to the entire ribosome. The transcriptome or proteome significant changes were transformed into binary input (Appendix A) where one equals significantly changed and zero not changed. The SciPy implementation of the Fisher’s exact test was used [150]. To counter the effects of multiple testing, the *p* values generated from Fisher’s exact test were adjusted via a strict, Bonferroni correction [88], and subsequently re-evaluated for significance.

### 4.10. Sequence Alignments

The coding sequence region of the nuclear encoded uL30_RPL7 gene RP family for the three taxa *Arabidopsis thaliana*, *Homo sapiens*, and *Saccharomyces cerevisiae* were aligned using the online Guidance2 Server [151,152,153]. The Bayesian inference analysis was performed in BEAST—v2.6.1 [154] and the parameters were assembled in Beauti, which is a part of the BEAST software package. The alignment was analyzed using a lognormal relaxed clock [155], a GTR + i model (as determined by the corrected AIC criteria in jModelTest 2.1.10 [156,157]—see Appendix A), and a uniformly distributed Yule process speciation tree [158]. Markov chain Monte Carlo (MCMC) was set to 10,000,000 generations [159]; sampling was performed every 1000 generations and the burn-in set to 10%. The MCMC trace files generated were visualized in Tracer v1.7.1 [160], which presented statistical ESS summaries over 300 (Appendix A). The TreeAnnotator tool from BEAST was used to combine all the log trees with a discarded burn-in of 10% into a single, maximum clade credibility tree, using common ancestor heights and a posterior probability of 0.5 [154]. FigTree [160] was used to visualize the final phylogenetic tree, the node labels were set to show the posterior probability values and the tree was re-rooted; *Saccharomyces cerevisiae* was set as the outgroup.

### 4.11. Software

Structural work was done using python scripts (Python programming language, RRID:SCR_008394) in a linux terminal through the enabler SmarTTY.

Fisher exact tests were executed in python 3: Jones, E., Oliphant, T., Peterson, P. & Others. SciPy.org. *SciPy: Open source scientific tools for Python2* (2001).

Structural visualization was enabled via PyMol [161]. (PyMOL, RRID:SCR_000305). PyMOL: The PyMOL Molecular Graphics System, Version 2.0 Schroedinger, LLC.

Transcriptome and proteome interactive data analyses were carried out in R [162,163]: R foundation for statistical computing, https://www.R-project.org/, accessed on 30 April 2021.

The following R packages were used (we report either a citation in the references or a link to the package website in this section):Ben Bolstad (2019). preprocessCore: A collection of pre-processing functions. R package version 1.46.0. Available online: https://github.com/bmbolstad/preprocessCore accessed on 30 April 2021.Ref. [164]. tidyverse: Easily Install and Load the ‘Tidyverse’. R package version 1.2.1.Ref. [165]. Reshaping Data with the reshape Package.Ref. [166]. stringr: Simple, Consistent Wrappers for Common String Operations. R package version 1.4.0.Ref. [133]. fitdistrplus: An R Package for Fitting Distributions.Ref. [167] ggplot2: Elegant Graphics for Data Analysis.Ref. [168] R package stringi: Character string processing facilities.Diethelm Wuertz, Tobias Setz and Yohan Chalabi (2017). timeSeries: Rmetrics—Financial Time Series Objects. R package version 3042.102. Available online: https://CRAN.R-project.org/package=timeSeries accessed on 30 April 2021.Ref. [169]. Complex heatmaps reveal patterns and correlations in multidimensional genomic data. Bioinformatics.Ref. [170]. Circlize implements and enhances circular visualization in R. Bioinformatics.Ref. [149] The igraph software package for complex network research.Ref. [171]. VennDiagram: Generate High-Resolution Venn and Euler Plots. R package version 1.6.20.Jan Graffelman (2013). calibrate: Calibration of Scatterplot and Biplot Axes. R package version 1.7.2. Available online: https://CRAN.R-project.org/package=calibrate accessed on 30 April 2021.Diethelm Wuertz, Tobias Setz and Yohan Chalabi (2017). fBasics: Rmetrics—Markets and Basic Statistics. R package version 3042.89. Available online: https://CRAN.R-project.org/package=fBasics accessed on 30 April 2021.

## 5. Conclusions

A ribosome biogenesis transcriptional response is triggered 1 day after plants experience cold suboptimal temperature. The response is followed by differential accumulation of RP transcripts and proteoforms 7 days after the initial cue. The outcome is altered RP stoichiometry in non-translational and low-oligomeric translational ribosomal fractions. The divergent ribosomal populations arising during cold present altered stoichiometry around the PET, both in Arabidopsis wild type and two independent *reil* double knock out genotypes. Thus, altered stoichiometry of the PET might be a typical cold response triggered during early biogenesis and could explain why REIL proteins are absolutely necessary only during cold to mature competent 60S subunits. In alignment with the former statement, uL30s are depleted from active translating polysomes in Arabidopsis during cold. In yeast, uL30 depletion causes PET-altered stoichiometry. Thus, supported by the functional roles and sequence alignments of uL30, we suggest that these two events are linked in the context of cold acclimation. After REIL action, the P-Stalk is assembled and altered stoichiometry is observed in the P-Stalk at the transcriptome level during early biogenesis. Further experiments are needed to verify the P-Stalk remodeling at the proteome level. We propose that REIL proteins could enable maturation of PET-remodeled LSUs that lead toward ribosome specialization supported in subsequent direct or indirect REIL actions upon the P-Stalk and ribosomal head substructures.

## Figures and Tables

**Figure 1 ijms-22-06160-f001:**
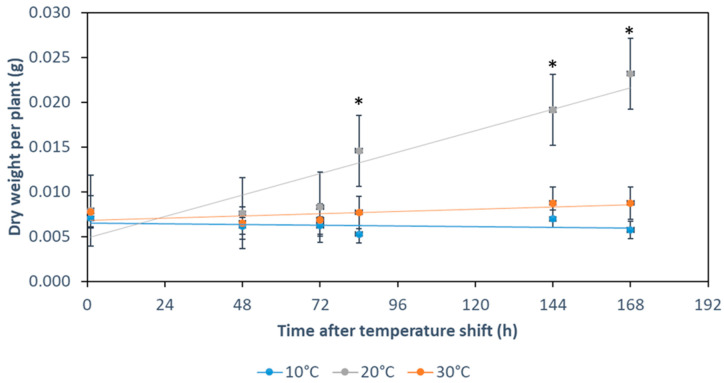
Arabidopsis Col-0 arrests biomass accumulation after shift to suboptimal temperatures. Plants were temperature shifted in the middle of the light phase at developmental stage 1.10, i.e., a rosette with 10 leaves with at least 1 mm in length, as defined by Boyes et al. [55]. Shift at 0 h from 20 °C to 30 °C (orange), to 10 °C (blue), or continuous 20 °C (grey). Note that control plants at 20 °C continued to accumulate dry weight during the first 7 days. Plants submitted to suboptimal temperature, either 10 °C or 30 °C, did not accumulate dry weight in the same period. The dry weight differences at and following 84 h of temperature acclimation were significant (asterisks, *t*-test significance *p* < 0.05). Means were calculated based on three biological replicates at each time point. The error bars correspond to the maximum standard deviation encountered across time points (Appendix A). Biological replicates were composed of three independent plants grown in hydroponic cultivation systems (Appendix A).

**Figure 2 ijms-22-06160-f002:**
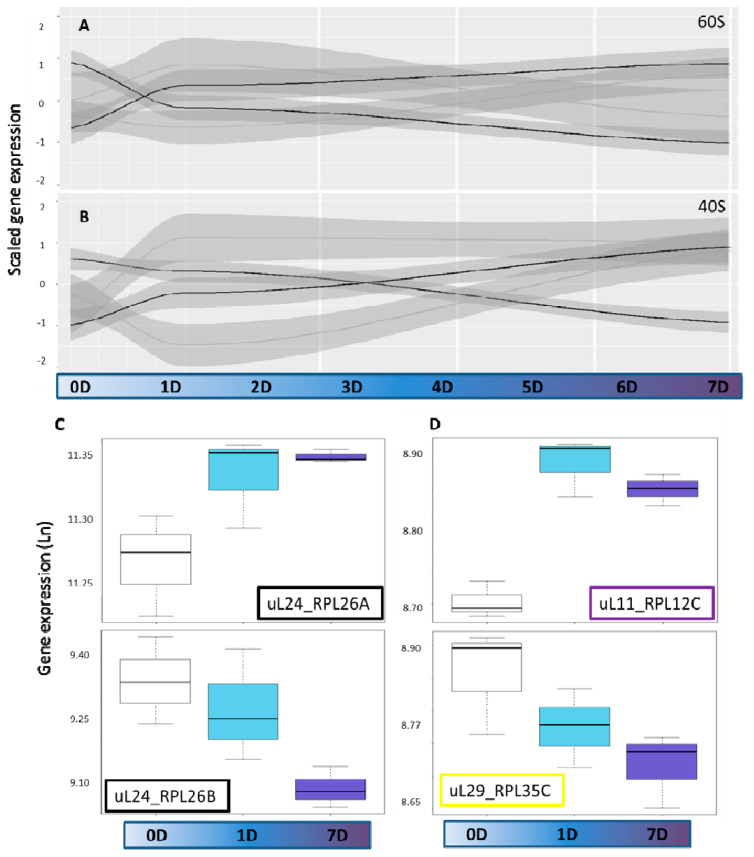
Changes of gene expression of cytosolic structural ribosome proteins in roots of *Arabidopsis thaliana* acclimating to suboptimal temperature. Arabidopsis wild type Col-0 plants grew at 20 °C to vegetative developmental stage 1.10 [55]. The transcriptome of whole root systems was profiled prior to a 10 °C cold shift (0D) and at 1 or 7 days after (1D or 7D) the shift. Replicates *n* = three independent biological replicates at each time point. Probe hybridization intensities are natural logarithm-transformed, quantile-normalized, and statistically tested applying a generalized linear model with details reported in the methods section using R functions compiled in the GitHub repository RandoDiStats (https://github.com/MSeidelFed/RandodiStats, accessed on 30 April 2021). Scaled differential intensities cluster into four expression trends (K-means clustering) and identify response groups of cytosolic RP genes (Appendix A). Cluster means, indicated by solid grays and standard error intervals, light gray underlay, reveal four expression trends in (**A**) 60S LSU and (**B**) 40S SSU. (**C**) Example of inversely regulated paralog transcripts from the RPL26 (uL24) ribosomal protein family. (**D**) Examples of up and downregulated transcripts of the uL11 and uL29 ribosomal protein families respectively. The remaining paralogs within these two exemplary families did not show significant changes in their abundances and thus are not shown in the figure. Note that (**C**,**D**) represent non-scaled natural logarithm-transformed probe hybridization values. Colored boxes of ribosomal protein names indicate the temperature dependent type of upregulation or downregulation in paralog expression patterns (see Figure 3).

**Figure 3 ijms-22-06160-f003:**
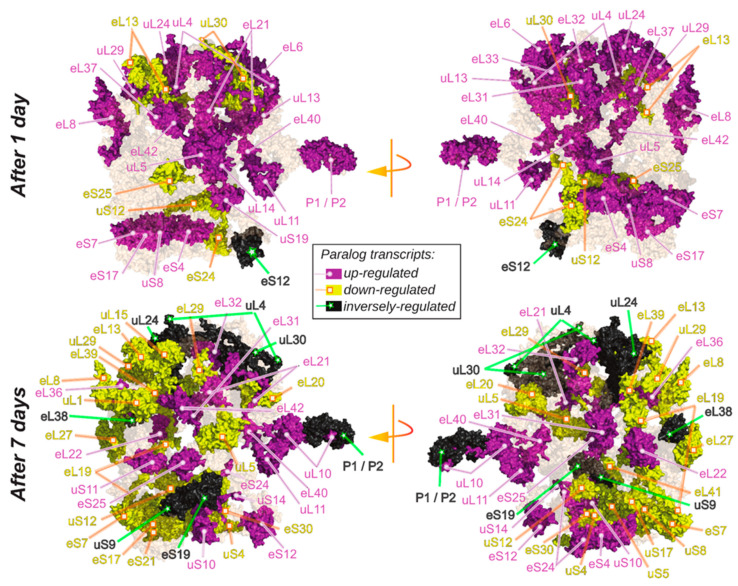
Differentially expressed ribosomal paralog genes from RP families of Arabidopsis thaliana root. Visualized data refers to significant differential hybridization of one or multiple probes per gene after 1 day (**top**) or 7 days (**bottom**) of cold exposure (Appendix A). Differential paralog expression is mapped onto a 3D rendering derived from a Cryo-EM model of the wheat 80S ribosome [59]. Statistical analyses are identical to those in Figure 2. Purple-colored proteins have increased transcript abundances, yellow decreased abundances, and black inversely regulated transcript abundances of a single or multiple paralogs of the same gene family. Note left and right parts of the figure rotate the model by 180°. The model does not contain the rRNA scaffold.

**Figure 4 ijms-22-06160-f004:**
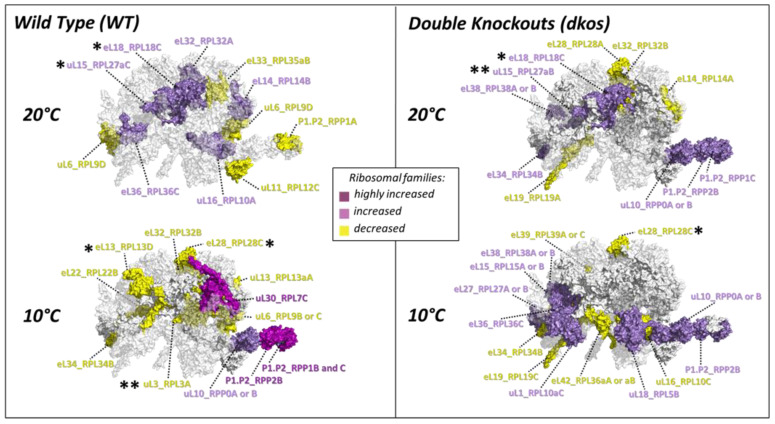
Substoichiometry of 60S proteome LSU-RPs 7 days after shifting from 20 °C to 10 °C in Arabidopsis thaliana Col-0 roots (**left panel**) and at the same time point in two double knockouts (*dkos*) genotypes of REIL proteins (**right panel**). Normalized protein abundances (Appendix A) were used to calculate Ln-transformed ratios between the 60S and the polysome fraction. Only RPs that appeared in all replicates across the ratio components were considered. Ratios were genotype and temperature-specific (Appendix A). The resulting RP paralog ratios were divided into four groups: dark-purple, high increase or higher abundance of one or more RP paralogs in polysomes (Ln-ratios < 2.0 and > 1.0); purple, increase or more abundance (Ln-ratios > 0.0 and ≤ 1.0); yellow, decrease or less abundance (Ln-ratios ≥ −1.0 and < 0.0); off-white, no change. The 60S-RP ratios that belonged to the highest and lowest ten magnitudes were colored into a 3D rendering of the 60S subunit. ***** Black asterisks indicate RP families with significant substoichiometric paralogs (GLM—*p* Values > 0.01 ^ < 0.05 = **; > 0.05 ^ < 0.1 = *) in polysomic compared to the 60S fractions with a single regressor, namely temperature as the sole experimental factor (*n* = 4; *dko*1, *dko*2, WT-DS1, WT-DS2). Only one significant change does not coincide with the largest fold changes and hence is not visualized (i.e., eL8_RPL7aB—see Appendix A).

**Figure 5 ijms-22-06160-f005:**
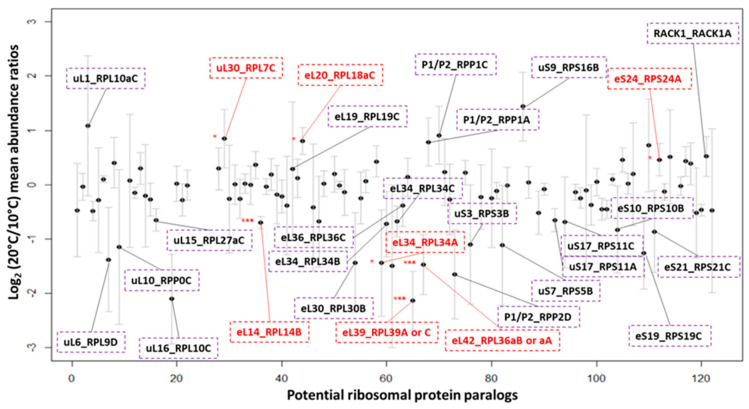
RP candidates of temperature-induced rearrangements in active translating ribosomes. Mean Log_2_ ratios of 20 °C/10 °C ribosomal protein abundances during cold acclimation in *Arabidopsis thaliana* roots across independent genotypes in polysomes (*n* = 4; *dko1*, *dko2*, WT-DS1, WT-DS2 outlined in Appendix A). The abundances were LFQ-normalized and corrected to the total amount of protein per ribosomal subunit in order to avoid relative subunit abundance to influence the results (Appendix A). The x-axes contains 123 ribosomal proteins in common between two independent shotgun proteomics runs. The y-axis contains the mean Log_2_ ratios of normalized abundances and the error bars represent the standard error. Only RPs with a Log_2_ ratio >0.5 or <−0.5 have been named in the plot. In order to account for RPs that were cold-specific in at least one replicate causing an -inf as a result of the ratio, -inf was replaced by −2 only for graphical purposes. This means that those values did not affect the statistical testing. Statistical testing (p Values < 0.01 = ***; > 0.01 ^ < 0.05 = **; >0.05 ^ < 0.1 = *) was done by fitting a GLM of the Gaussian family after testing the distribution of protein abundances using R functions compiled in the GitHub repository RandoDiStats (https://github.com/MSeidelFed/RandodiStats, accessed on 30 April 2021) and realizing that after normalization proteins in our datasets approximate a normal distribution (Appendix A).

**Figure 6 ijms-22-06160-f006:**
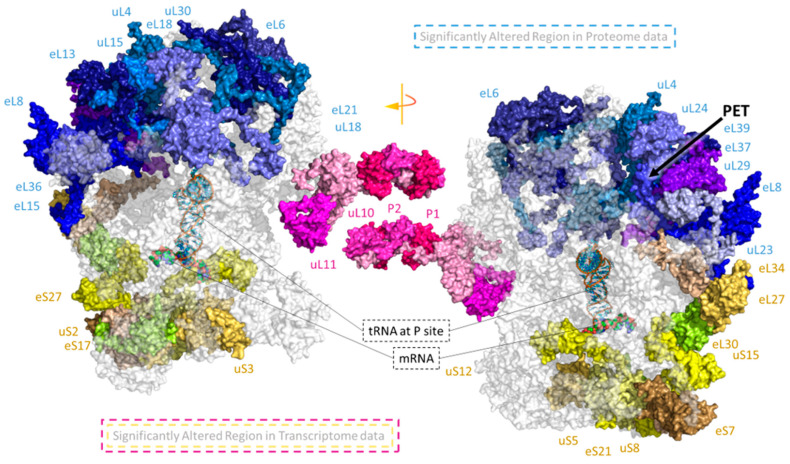
Significantly altered ribosomal regions during cold acclimation in *Arabidopsis thaliana* roots measured in three independent genotypes and four biological replicates in total. The figure is a visualization of the random walk sampling results and tested regions of interest detailed in the GitHub repository COSNet_i_ (https://github.com/MSeidelFed/COSNet_i, accessed on 30 April 2021). Sampled regions with a significantly increased proportion of RP changes as compared to the entire ribosome are depicted based on testing of the transcriptomic (shades of magenta and yellow) and proteomic (shades of blue) datasets. The binary input for the tests has been compiled in Appendix A. Additionally, several regions had significant results at a threshold *p* value < 0.05 before the multiple testing correction and are reported in Appendix A. Note that at the transcript level one of the significantly altered regions co-localizes with the P site where tRNA binds as well as where the decoding of mRNA occurs. At the proteomic level, the significances co-localized with the polypeptide exit tunnel (PET) depicted with a solid arrow. Both ribosomal images represent a 180° rotated mirror image of each other.

**Figure 7 ijms-22-06160-f007:**
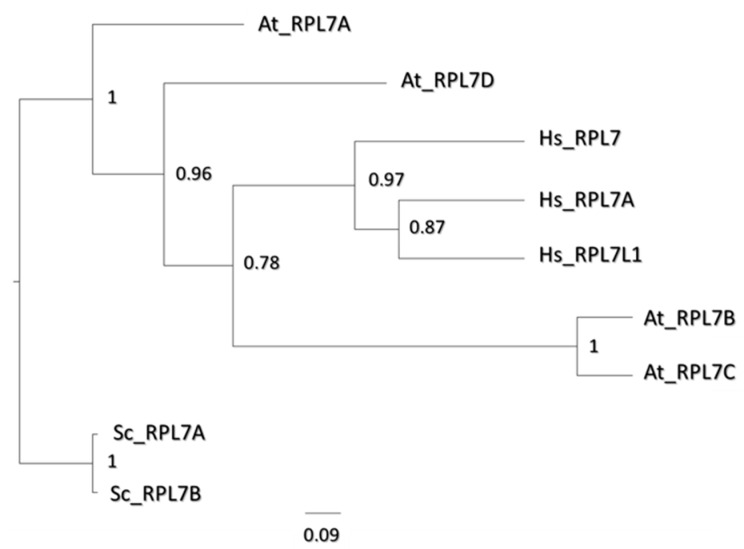
Phylogenetic tree resulting from the Bayesian analysis of the uL30 ribosomal protein gene family found in *Arabidopsis thaliana*, *Homo sapiens,* and *Saccharomyces cerevisiae*. Node values represent posterior probabilities calculated through Bayesian confidence methods (Appendix A). Branch lengths represent substitutions for each branch, where the bottom scale represents the length of a 9% nucleotide substitution rate per site (i.e., number of changes per 100 nucleotides) according to a hypothetical common ancestor.

**Table 1 ijms-22-06160-t001:** Paralogs of RP families that showed altered stoichiometry in 60S and polysome fractions.

Temperature	Genotypes	More Abundant in 60S	More Abundant in Polysomes
20 °C	Genotype-independent *	uL15_RPL27aB, uL15_RPL27aC, eL18_RPL18C	
10 °C			uL3_RPL3A, eL28_RPL28C, eL13_RPL13D
20 °C	WT	eL36_RPL36C, eL18_RPL18C, eL32_RPL32A, uL15_RPL27aC, eL14_RPL14B, uL16_RPL10A	P1.P2_RPP1A, uL11_RPL12C, uL6_RPL9D, eL27_RPL27A or B, eL33_RPL35aB
	*dkos*	eL34_RPL34B, eL38_RPL38A or B, uL10_RPP0A or B, P1.P2_RPP1C, P1.P2_RPP2B	eL19_RPL19A, eL14_RPL14A, eL28_RPL28A, eL32_RPL32B
10 °C	WT	P1.P2_RPP2B, uL10_RPP0A or B, uL30_RPL7C, P1.P2_RPP1C, P1.P2_RPP1B, P1.P2_RPP2A	eL22_RPL22B, eL32_RPL32B, uL13_RPL13aA, uL6_RPL9B or C
	*dkos*	eL15_RPL15A or B, P1.P2_RPP2A, uL18_RPL5B, eL36_RPL36C, eL38_RPL38A or B, uL10_RPP0A or B, uL1_RPL10aC, eL27_RPL27A or B	uL16_RPL10C, eL42_RPL36aA or aB, eL39_RPL39A or C, eL19_RPL19C

* Signals the significant changes, i.e., shared among genotypes, which are likely to arise during early biogenesis.

**Table 2 ijms-22-06160-t002:** Paralog abundance changes that imply significantly modulated regions during cold acclimation in plants. Different sources of omics data were interpreted in their structural context using the procedures detailed in the GitHub repository COSNet_i_ (https://github.com/MSeidelFed/COSNet_i, accessed on 30 April 2021). The data tested were transcriptome (1T) inversely regulated, (2T) inverse and upregulated, (3T) inverse, down, and upregulated as detailed in Figure 3. Proteome (1P) 60S to polysome ratios, (2P) polysome or (3P) both instances as detailed in Figure 4 and Figure 5. Multiple entries per test indicate that several regions tested significant according to a Fisher exact test comparing the proportion of RP significances within selected regions to the total amount of significances among RPs.

Regions	Ribosomal Protein Families	Region Identifiers	*p* Values (Fisher)	Q Values (Bonferroni)
Transcriptomics		4v7e_d_t_8_IN20_WL10		
1T—A	eL39 eL37 uL24 uL23 uL4 uL29 eL8 uL30 eL27 eL30 eL34	LSU-PET	0.07	0.80
1T—A	uL11 P1 P2 uL10	LSU-P-Stalk	0.05	0.51
2T—B	***uS15 uS12 uS8 uS3 eS17 uS2 eS7 eS21 eS27 uS5 eL27 eL30 eL34***	**SSU-LSU-RibosomalHead.1**	***0.00***	***0.02***
2T—B	***uS15 uS12 eS17 eL19 eS7 uS2 uS17 eS21 eS27 uS5 uS8***	**SSU-LSU-RibosomalHead.2**	***0.00***	***0.00***
2T—B	uL11 P1 P2 uL10	LSU-P-Stalk	0.00	0.05
3T—C	uL13 eL24 eS8 uL14 uL3	SSU-LSU-uL3 region	0.01	0.18
3T—C	uL11 P1 P2 uL10	LSU-P-Stalk	0.02	0.36
Proteomics		4v7e_d_t_8_IN20_WL10		
1P—D	eL13 eL15 uS7 uL15 eL18 uL29 eS25 uL23 uL5 eL36 eL42 eL8 uL30 uS13 eL21 uS11	SSU-LSU-uL30 region	0.03	0.42
1P—D	uL15 uL13 eL24 eS8 uL14 uL3	SSU-LSU-uL3 region	0.03	0.39
3P—E	***eL13 eL15 eL18 uL15 eL39 eL37 uL4 uL24 eL6 uL23 uL29 eL36 eL8 uL30 eL21 uL18***	**LSU-PET**	***0.00***	***0.04***
3P—E	eL32 eL13 eL18 uL15 eL33 uL4 eL14 eL20 eL6 uL13 uL30 eL21 uL3 eL28	LSU-uL30-uL3 region	0.00	0.05

IN: iteration number, WL: Walking length, d_t_: distance threshold in Å, **bold**: significant regions after Bonferroni correction.

## Data Availability

The expression data sets are available from the Gene Expression Omnibus (https://www.ncbi.nlm.nih.gov/geo/, accessed on 30 April 2021) through accession GSE144916. The mass spectrometry proteomics data have been previously deposited to the ProteomeXchange Consortium via the PRIDE partner repository [142] with the dataset identifier PXD016292.

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
