# Peer review of "Spatially Enriched Paralog Rearrangements Argue Functionally Diverse Ribosomes Arise during Cold Acclimation in Arabidopsis"

_ijms, 2021, doi:10.3390/ijms22116160_

Round 1
Reviewer 1 Report
The paper by Martinez-Seidel et al describes how plant (arabidopsis) ribosomes change in composition and structure during cold adaptation. It provides a significant amount of information on the process of ribosome biogenesis under stressful circumstances and sheds precise and rigorous light on the recent concept of ribosome heterogeneity with robust and novel results. This paper is overall well written, the introduction is rich in information but I feel it is necessary to discuss the following points before this article is published
1. The process of cold-acclimation is a complex one that must also take into account the effect of lower T° on the physical properties of r-proteins such as their folding and stability. Before entering into the ribosomal particles, the r-proteins must fold and there are often co-folding phenomena between the proteins and the rRNA. These processes are likely influenced by the T°. The authors should therefore discuss the general point of protein stability and cell abundance (PMID: 30150386) and the cold-denaturation effect on proteins : PMID: 2225910, PMID: 25198426, PMID: 27513457, PMID: 19616532). In fact, ribosomal proteins are particular in that most of them have disordered segments in the absence of their partners. Some of them display particular behaviours according to the temperature (PMID: 23461364, PMID: 20225821). It seems to me important to discuss this point : it could also help to elucidate why some r-proteins (depending on their sequence/structure and their different properties depending on the temperature) are expressed differently depending on the T°.
2. Where will the additional r-proteins be located in the differentiated ribosomes? It would be interesting to draw inspiration from the work on archaeal ribosomes, which show that the same r-proteins can occupy different sites (PMID: 23222135). An analogous process could take place when there are too many proteins.
The authors could also consider the role of disordered rRNAs (expansion segments), which could play a role in the binding of supernumerary r-proteins. [Please, also quote the first high-resolution structure of eukaryotic ribosome that helped to decipher in detail all the interactions between its components (PMID: 22096102)]
3. Recent studies have shown that r-proteins form complex “neuron-like” networks whose mathematical properties have evolved to optimize information transfer between functional sites (PTC, tunnel, tRNAs sites, etc) (PMID: 27225526, PMID: 31207893, PMID: 33436806). In these papers it is predicted that ribosome heterogeneity (especially of r-proteins) can affect the connectivity in these networks and modify information transfer between the functional centres. I recommend discussing how different expressions and stoichiometry of some of r-proteins in cold-acclimated “heterogeneous” ribosomes alter protein networks and may influence their function at low temperature.
Author Response
The paper by Martinez-Seidel et al describes how plant (arabidopsis) ribosomes change in composition and structure during cold adaptation. It provides a significant amount of information on the process of ribosome biogenesis under stressful circumstances and sheds precise and rigorous light on the recent concept of ribosome heterogeneity with robust and novel results. This paper is overall well written, the introduction is rich in information but I feel it is necessary to discuss the following points before this article is published
- The process of cold-acclimation is a complex one that must also take into account the effect of lower T° on the physical properties of r-proteins such as their folding and stability. Before entering into the ribosomal particles, the r-proteins must fold and there are often co-folding phenomena between the proteins and the rRNA. These processes are likely influenced by the T°. The authors should therefore discuss the general point of protein stability and cell abundance (PMID: 30150386) and the cold-denaturation effect on proteins : PMID: 2225910, PMID: 25198426, PMID: 27513457, PMID: 19616532). In fact, ribosomal proteins are particular in that most of them have disordered segments in the absence of their partners. Some of them display particular behaviours according to the temperature (PMID: 23461364, PMID: 20225821). It seems to me important to discuss this point : it could also help to elucidate why some r-proteins (depending on their sequence/structure and their different properties depending on the temperature) are expressed differently depending on the T°.
R/ We thank the reviewer for the nice and accurate comment. We have used the information to build a new section in our discussion in order to further elaborate on our own results. Line 816 (discussion):
“
3.4. Cold Dynamics of Ribosomal Protein Assembly
Assembly and specific interactions of different RP or RP paralogs may be a product of the effect of lower temperatures to protein stability and folding states. Agozzino and Dill [103] described a model for the sequence adaptation of proteins according to temperature changes and argued that the least stable proteins are the ones that adapt faster to temperature changes and help the organism adapt faster. In this process, chaperones and their properties are essential as capacitors of cellular evolution. One process that can alter folding states of globular proteins is cold denaturation [104–107]. Besides heat, cold can denature proteins due to the disturbance of the forms of noncovalent bonding that are responsible for the folding state at physiological conditions. Thus, cold denaturation is essentially dependent of the protein structure and the hydrophobic effect [106]. Due to different amino acid composition and structural features, the temperature in which cold triggers denaturation can vary for different proteins.
At 10°C in our experiments, cold denaturation could account for the spatially changed PET region in both transcript and proteome data (Figure 6, Table 2) and the choice of assembling different RP or RP paralogs. Many RNA-binding proteins have linkers as intrinsically disordered regions (IDRs) in their natural native state, which confer flexibility and plasticity to the RNA-binding domains [108]. In RPs, IDRs are disordered extensions that can stretch to different parts of the subunits. Extensions can penetrate the core of subunits, transition to a more ordered structure and assist rRNA folding, and different categories of extensions may have distinct functions in assembly stages [109–111]. Moreover, many eukaryotic RPs disordered extensions interact with the rRNA expansion segments (ES) around the peptide exit tunnel (PET) [83], and from the P-stalk to the L1 side. ES are present in the surface of both subunits, but in the 60S they are more abundant and form a nearly continuous ring around the PET. It may be that cold structurally modulated RP paralog extensions can lead to different interactions with rRNA helices and ES due to different folding states, leading to heterogeneous subpopulations of subunits and ultimately ribosomal complexes.
”
- Where will the additional r-proteins be located in the differentiated ribosomes? It would be interesting to draw inspiration from the work on archaeal ribosomes, which show that the same r-proteins can occupy different sites (PMID: 23222135). An analogous process could take place when there are too many proteins.
- The authors could also consider the role of disordered rRNAs (expansion segments), which could play a role in the binding of supernumerary r-proteins. [Please, also quote the first high-resolution structure of eukaryotic ribosome that helped to decipher in detail all the interactions between its components (PMID: 22096102)]
R/ We have drawn inspiration from the suggested paper to argue that the canonical stoichiometry in plant non-translational 60S complexes, which is a surplus of RPs, may be deeply interconnected to the idea of supernumerary RPs and promiscuous binding to different parts of the ribosomal complexes. This is discussed in the context of rRNA expansion segment logarithmic accumulation in metazoan ribosomes, which could serve as the scaffold for these binding events.
R1/ Line 791 (discussion): “Importantly, the intrinsic increased abundances across several RPs in the 60S population before the temperature shift indicates a surplus of proteins bound to the non-translational 60S subunits. RPs can exhibit promiscuous binding in archaeal ribosomes with some proteins being present at more than one location per 50S ribosome [94]. Moreover, metazoan cytosolic ribosomes acquired novel expansion segments (ES) logarithmically over the past two billion years as compared to archaeal and bacterial ribosomes [95,96]. Metazoan RPs have also diversified, increased in number and in parts diverged [97–101]. Thus, it is likely that the availability of novel rRNA expansion segments and the diversification of RPs enhances promiscuous binding of multiple RP or RP paralog copies per ribosomal particle in metazoans and specially also in plant ribosomes. Moreover, storing these “RP abundant” complexes in a non-translational fraction might be a on the run strategy to rapidly tune the ribosomal network and meet translational needs. Consequently, the surplus of RPs in 60S subunits supports one of our previous notions, i.e., “that Arabidopsis may buffer fluctuating translation by pre‑existing non‑translating ribosomes before de novo synthesis meets temperature‑induced demands” [38].”
R2/ we cited the first high-resolution structure of the eukaryotic ribosome. Line 735 (discussion): “Selecting ribosomal regions and studying the interactions between all ribosomal components was initially enhanced by the first high-resolution structure of a eukaryotic ribosome that helped to decipher in detail all the interactions [83]. This type of information allowed us to define coherent ribosomal regions, which is the most influential step towards finding spatial enrichments.”
- Recent studies have shown that r-proteins form complex “neuron-like” networks whose mathematical properties have evolved to optimize information transfer between functional sites (PTC, tunnel, tRNAs sites, etc) (PMID: 27225526, PMID: 31207893, PMID: 33436806). In these papers it is predicted that ribosome heterogeneity (especially of r-proteins) can affect the connectivity in these networks and modify information transfer between the functional centres. I recommend discussing how different expressions and stoichiometry of some of r-proteins in cold-acclimated “heterogeneous” ribosomes alter protein networks and may influence their function at low temperature.
R/ We thank the reviewer for the insightful suggestion and consider it as a very relevant point of discussion. We have introduced editions in the discussion section highlighting the relevance of RP networks in the information transfer process and discussing the implications of altered RP networks during cold acclimation in plant ribosomes. We connected in this way in a more efficient manner to the subsequent section in the discussion.
R1/ Line 753 (discussion): “Spatial adjustments of the ribosomal proteome may be at the core of ribosome specialization. The functional centers of ribosomes actively communicate to perform ribosomal functions. The communication among functional centers is optimized by the coevolution of a non-random RP network [88]. These RP graphs harbor specific neuron-like properties that led to the realization of RPs being instrumental in the process of information transfer across ribosomal complexes [89,90]. For example, mutating specific amino acid residues from L11 in yeast leads to impaired inter-subunit communication, which in turn causes structural alterations in 40S and 60S rRNA [91], high-lighting the large scale of information flow within the network. Similarly, L3 in yeast plays a role in the coordination of the elongation cycle by communicating the tRNA site status to the elongation factor binding region and the peptidyltransferase center [92]. Thus, small changes in RPs can lead to vast rearrangements in the RP network and adjustment of ribosomal function. We used the plant RP network to define coherent ribosomal regions and used those regions to interpret our omics data in their structural context. We found that altered stoichiometry in specific RP and RP paralogs is correlated to modulated ribosomal substructures during cold acclimation in plants. Suggesting for example that uL30 acts as a communication hub that restructures the ribosome during cold. Moreover, we argue that after stopping ribosome biogenesis during cold, the modulation of these substructures is likely to occur in the early nucleolar biogenesis as a response to maintain protein synthesis during acclimation.”
Reviewer 2 Report
The authors performed transcriptome and mass spectrometry analysis during cold acclimation to evaluate the expression level of the ribosomal protein (RP) genes and the difference in composition between the 60S and polysome. Based on those results, the authors analyzed the distribution pattern of ribosomal proteins among the paralogs during the cold response. As a result of the analyses, the authors found a ribosome region that tends to change during cold acclimation. It could be a starting basis for finding the significance of the gene duplication in plant ribosomes in response to low temperature. However, there are some inadequacies in the experiment’s validity, how the figures and tables are presented, and the description of the results, which are pointed out below.
Major points
Neither the transcriptome nor the MS analysis has positive control data to show that the cold treatment works.
There is no data to show that the polysome fractionation experiment is working successfully.
Overall, the descriptions are verbose, and it isn't easy to understand what the results were. It would be better to use more graphs to show the results.
In many cases, only the classification of whether a change has occurred or not is described, so it is not clear from the text how much of a change low temperature treatment has resulted in. For example, if the transcript levels of RPXA and RPXB were “1:1000” under normal conditions and changed to “3:1000” in response to low temperature, it should not be considered that a significant effect occurred, but in this manuscript, isn't RPXA considered to have increased significantly in such a case? In my opinion, it is necessary to describe the degree of relative changes between paralogs.
Minor points
A huge number of reference errors.
Fig. 2A. The number of treatment days is 0, 1, and 7, but each is equally spaced horizontally, not acceptable.
Fig. 2D. It is described that only RPL12C showed this characteristic expression pattern among the paralogs. If that is the case, the results of RPL12A/B should be presented together.
Fig. 2C and 2D. How did the authors make a box plot with only three samples?
Line259-263. It is necessary to describe which description corresponds to which ClusterNo. in FigS3. Such incomprehensible descriptions in reading the figures are frequently seen.
Fig. 3. I can hardly recognize the color of the letters in each RP, and I can't distinguish where the dotted line ends.
Line315-318. It is described that the RPs of P-stalk and PET showed similar characteristic expression patterns, but the name of each RP should be clearly indicated. Also, it should be easy to understand whether all the RPs showed the same expression pattern or only some of them.
Fig. 4. I can't read the letters because of the inappropriate yellow color.
Fig. 4 legend. About p-value, *** doesn’t exist. Additionally, does the 0.1 to 0.05 range (*) means really statistically significant?
Fig. 4. Is the dKO result shown on the right DKO1 or DKO2?
Fig.4. It is stated that the RPs detected in all the samples were evaluated. Does this mean that all the RPs that were not colored had samples that were not detected? Or does it include "detected in all samples but no change"? In the latter case, both RPs should be described separately.
Table 1. Wouldn't it be more appropriate to use "more abundant" instead of "increased"?
Table 2. Not enough explanation. Regions?? 1T,2T,3T,1P,3P?? A,B,C,D,E?? P-value and Q-value of what compared to what??
Author Response
- The authors performed transcriptome and mass spectrometry analysis during cold acclimation to evaluate the expression level of the ribosomal protein (RP) genes and the difference in composition between the 60S and polysome. Based on those results, the authors analyzed the distribution pattern of ribosomal proteins among the paralogs during the cold response. As a result of the analyses, the authors found a ribosome region that tends to change during cold acclimation. It could be a starting basis for finding the significance of the gene duplication in plant ribosomes in response to low temperature. However, there are some inadequacies in the experiment’s validity, how the figures and tables are presented, and the description of the results, which are pointed out below.
R/ We thank the reviewer for their remarks and questions, we try to answer and implement thorough editions as much as possible across the introduction, results and discussion sections.
Major points
- Neither the transcriptome nor the MS analysis has positive control data to show that the cold treatment works.
R/ We appreciate the comment and we now see the importance for further explanation. The cold treatment used in our studies (temperature-shift from 20 to 10°C) and its effect in our genotypes has been widely characterized. In more detail, early in Schmidt et al. 2013 (39) it was discovered that reil genotypes had a cold phenotype at 10°C. This was reiterated later in Beine-Golovchuk et al. 2018 (34). Accordingly in section “2.1. Early Temperature Acclimation Effects on Plant Growth”, we cite their results (growth and development rate decrease in terms of leaf number and rosette area), adding in this work the lack of dry weight accumulation. In Beine-Golovchuk et al. 2018 (34) classical cold responses at the transcriptome level like that of the DREB/CBF regulon were characterized. These cold transcriptome links were confirmed and enhanced by a different group using the same treatments, i.e., Yu et al. 2020 (43). Thus having already introduced over the years many publications detailing the cold responsiveness of our system and genotypes, we avoided yet another characterization of the system in order to delve into the more specific aspect of ribosome remodeling.
R1/ All these details had been previously written in the introduction in paragraph 3. However, as details were evidently lacking and following the reviewer legitimate inquiry we now state that the cold system used in this work has been widely characterized and shown to cause a classical plant cold response such as the DREB/CBF regulon modulation. To reflect that, we have added the following sentence to paragraph 3 in the introduction (Line 133): “In our current study, we used the same cold stress treatments previously characterized to trigger a classical plant cold response [34,39], in order to delve into the more specific aspect of translational reprogramming.”.
- There is no data to show that the polysome fractionation experiment is working successfully.
R/ We thank the reviewer for this comment, and we agree that the data had not been compiled in the manuscript, which beyond method should show reproducibility across samples. The main reason to not compile the data before was that the polysome fractionation was performed according to previous works of Cheong et al. 2021 (38) and Firmino et al. 2020 (49) as is described in section 4.6.Cytosolic Ribosomal Proteome Preparation. Nonetheless and in light of the reviewer´s very relevant request, we have now added Figure S4 that shows original exemplary ribosomal profiles of our own experiment. Additionally, we have cross-referenced Figure S4 in line 388 in the results section “Cytosolic Ribosomal Proteome Reprogramming”.
- Overall, the descriptions are verbose, and it isn't easy to understand what the results were. It would be better to use more graphs to show the results.
R/ We thank the reviewer for this point. Consequently, we have thoroughly reviewed all the results section and edited to remove any source of redundant speech that may be considered verbose. Additionally, we edited all the minor points suggested by the reviewer and we thank the reviewer for those points, which enhanced the readability and interpretability of the results section. Regarding increasing the number of graphs, we added the ribosome profile graph in order to present succinctly the reproducibility of our ribosome isolation methodology. Additionally, in order to prevent an overly complicated description of the results we did add Tables 1 & 2 in the previously submitted version of the manuscript. These tables compile the most important results of the manuscript. However, beyond these two tables we feel that already with seven figures and nine supplemental figures our manuscript is quite saturated with graphical abstractions covering the full scope of the results. That being said we are open to any specific advice for graphs and / or extra figures on behalf of the reviewer.
R1/ editions were made to adjust the results paragraph after Figure 4 (Line 442), Table 1 (Line 465), Figure 5 (Line 504) and Figure 6 (Line 583) to implement the reviewer suggestions, reduce redundant speech and deliver a clearer message. Additionally, editions and additions were made in the discussion to enhance our results in Line 753 to outline the relevance of ribosomal protein networks as sources of information flow within the complex. Line 735 to cite the first high-resolution ribosome structure that enabled elucidation of the interactions across all ribosomal components. Line 791 to discuss the relevance of having an intrinsic surplus of RPs in the canonical stoichiometry at control conditions in the context of relevant literature. Line 816 we implemented a new discussion section to contextualize our results to the dynamics of cold RP assembly.
- In many cases, only the classification of whether a change has occurred or not is described, so it is not clear from the text how much of a change low temperature treatment has resulted in. For example, if the transcript levels of RPXA and RPXB were “1:1000” under normal conditions and changed to “3:1000” in response to low temperature, it should not be considered that a significant effect occurred, but in this manuscript, isn't RPXA considered to have increased significantly in such a case? In my opinion, it is necessary to describe the degree of relative changes between paralogs.
R/ We agree with the reviewer that the raised point is very relevant. However, we argue that our own system to interrogate plant ribosomes needs to be analyzed not from a transcript fold change perspective but from a statistical point of view. To elaborate, as stated in line 235 “we analyze bulk root tissue that pre-formed at optimized temperature and subsequently acclimated to reduced temperature.”. Thus, we are dealing with a large cytosolic ribosomal transcriptome proportion that pre-formed at optimized temperature and is not necessarily responding immediately to cold. Moreover, while the pre-existing ribosomal populations are replaced, many ribosomal protein transcripts may exhibit large fold changes associated with high variability due in part to stochastic degradation. Thus, to filter out the high variance associated with these events we needed to apply a statistical test to firmly state that the increased or reduced transcript abundances even though might not exhibit the largest fold changes are reproducible with low standard deviations and variances. Thus, we argue that those statistically significant changes are more likely to carry the biological information about reprogramming of the cytosolic ribosomal transcriptome upon a cold cue. Finally, the accuracy of our choice is also reflected by the fact that the substructures selected as modulated at the transcriptome level during cold are related to what is known in literature about reil factors.
Minor points
- A huge number of reference errors.
R/ We checked all references and cross-references and corrected where necessary.
- 2A. The number of treatment days is 0, 1, and 7, but each is equally spaced horizontally, not acceptable.
R/ We thank the reviewer for the nice and necessary suggestion. We have now modified the time lapse in A and B panels of our figure to reflect the whole 7D lapse of duration.
- 2D. It is described that only RPL12C showed this characteristic expression pattern among the paralogs. If that is the case, the results of RPL12A/B should be presented together.
R/ We understand the suggestion of the reviewer and given that the size of the figure is quite big already we decided to modify the legend in order to precisely reflect that for instance paralogs A and B of RPL12 did not undergo significant abundance changes. Thus, the boxes in a boxplot across time would look flat with the statistical quartiles heavily overlapping each other. In consequence we feel that the modification to the legend suffice to send the message across. The part of the legend that deals with these exemplary families (D) now reads (Line 302): “(D) Examples of up and downregulated transcripts of the uL11 and uL29 ribosomal protein families respectively. The remaining paralogs within these two exemplary families did not show significant changes in their abundances and thus are not shown in the figure.”.
- 2C and 2D. How did the authors make a box plot with only three samples?
R/ We refer the reviewer to Table S2. This table contains three replicates per condition. That is three replicates for WT day 0, three replicates for WT day 1 and three replicates for WT day 7. Thus, each boxplot is made based on three replicates per condition, which is statistically valid because it allows calculation of all the relevant boxplot stats.
- Line259-263. It is necessary to describe which description corresponds to which ClusterNo. in FigS3. Such incomprehensible descriptions in reading the figures are frequently seen.
R/ We implemented the correction, identifying the cluster No. from Figure S3 within brackets in the text starting at Line 263.
- 3. I can hardly recognize the color of the letters in each RP, and I can't distinguish where the dotted line ends.
R/ We changed the font color in order to be able to distinguish between the three conditions and we added bright spots at the end of the dotted lines in order to be clear about the identity of each RP. The dotted lines have been colored as well to further distinguish the three conditions.
- Line315-318. It is described that the RPs of P-stalk and PET showed similar characteristic expression patterns, but the name of each RP should be clearly indicated. Also, it should be easy to understand whether all the RPs showed the same expression pattern or only some of them.
R/ We fully agree with this comment. We have added the following sentence in order to clarify all the points raised by the reviewer (Line 375): “Only a subset of RP transcripts were differentially regulated during cold. The identity of those transcripts as well as the outcome of a statistical test tailored to evaluate their spatial relationship in the ribosome structure are found in the results section 2.6 “Spatially Constrained Cold Triggered Ribosome Heterogeneity”.”
R1/ To answer the question of the reviewer, in the cross-referenced section we provide Table S6, which as detailed contains the individual paralog identity and the information of which paralogs are actually modulated at the transcript level.
- 4. I can't read the letters because of the inappropriate yellow color.
R/ We modified the yellow color of the fonts to a darker shade that enhances readability.
- 4 legend. About p-value, *** doesn’t exist. Additionally, does the 0.1 to 0.05 range (*) means really statistically significant?
R/ We have deleted *** from the legend. Regarding the single stars * featuring p-values between 0.05 and 0.1, this is the consensus among statistical languages, i.e., p-values between 0.05 and 0.1 are always signaled with one star in the GLM implementations. For instance in the R statistical language. Moreover, we consider these as significant given the rank of responsiveness of RPs in our system and thus we treat these proteins as candidate RPs that could control and / or be involved in triggered ribosome heterogeneity, building new hypothesis for future research. This holds special relevance if we consider that in our dataset three independent genotypes converge to characterize the potential early responsive candidates.
- Fig. 4. Is the dKO result shown on the right DKO1 or DKO2?
R/ We describe in the legend that it means the 2 dkos combined. Nevertheless, to prevent confusion we now extend the title of the right panel from “double knockout (dko)” to “double knockouts (dkos)”.
- 4. It is stated that the RPs detected in all the samples were evaluated. Does this mean that all the RPs that were not colored had samples that were not detected? Or does it include "detected in all samples but no change"? In the latter case, both RPs should be described separately.
R/ We think there is enough information already in the text to explain these questions and apologize to the reviewer in advance if they should think otherwise. To elaborate, in section 2.4. Substoichiometry in Non-Translating Versus Translating Ribosome Complexes line 408 we write “We tested whether RPs ubiquitously present in all samples (i.e., 66 RPs outlined in Table S4C) were significantly substoichiometric during cold.” In the legend of the figure, we write that “only RPs that appeared in all replicates (…) were considered.
R1/ Thus to answer the reviewer questions, the RPs that are not colored were either not detected in all samples or were detected but not changed. To understand which RPs or RP paralogs belong to which group we have provided Table S4.
R2/ We are happy to provide more information shall the reviewer find it necessary
- Table 1. Wouldn't it be more appropriate to use "more abundant" instead of "increased"?
R/ We agree. We changed “increased” to “more abundant”.
- Table 2. Not enough explanation. Regions?? 1T,2T,3T,1P,3P?? A,B,C,D,E?? P-value and Q-value of what compared to what??
R/ We thank the reviewer for the comment. In order to enhance the interpretability in the table we have extended the table legend to read (Line 602) “Table 2. Paralog abundance changes that imply significantly modulated regions during cold acclimation in plants. Different sources of omics data were interpreted in their structural context using the procedures detailed in the GitHub repositoriy COSNeti (https://github.com/MSeidelFed/COSNet_i). The data tested were transcriptome (1T) inversely regulated, (2T) inverse and upregulated, (3T) inverse, down and upregulated as detailed in Figure 3. Proteome (1P) 60S to polysome ratios, (2P) polysome or (3P) both instances as detailed in Figure 4 & 5. Multiple entries per test indicate that several regions tested significant according to a Fisher exact test comparing the proportion of RP significances within selected regions to the total amount of significances among RPs.”.
Round 2
Reviewer 1 Report
The new version is fine and is now acceptable for publication
(please use the universal nomenclature for naming ribosomal proteins in the new paragraphs)